# COALA: Convex Optimization for Alignment and Preference Learning on a Single GPU

## Abstract

Fine-tuning large language models (LLMs) to align with human preferences has driven the success of systems such as ChatGPT and Gemini. However, methods like Reinforcement Learning from Human Feedback (RLHF) remain computationally expensive and complex. Direct Preference Optimization (DPO) offers a simpler alternative but has limitations such as inconsistent ranking accuracy, reliance on a frozen reference model, and high dependence on expensive GPU resources. We introduce *COALA*: a novel lightweight algorithm with theoretical guarantees which leverages the convex optimization reformulation of neural networks (cvxNN). By exploiting the expressiveness of cvxNN, COALA eliminates the need for a reference model and obtains significant reduction in both training time and VRAM consumption, thus enabling efficient training on a single GPU. Experiments across three datasets—including a 23,228-sample synthetic educational feedback dataset—and five models (including LLaMA-8B) demonstrate that COALA outperforms traditional preference alignment methods such as DPO and ORPO. COALA exhibits stable, monotonically increasing rewards and reaches peak margins in significantly shorter time in comparison to competitors. To the best of our knowledge, this is the first time convex optimization has been effectively applied to preference fine-tuning of LLMs.

## 1 Introduction

Large Language Models (LLMs) have been trained on increasing amounts of data to capture semantic language patterns and scale to solve more complex problems. The main paradigm combines pre-training and fine-tuning LLMs to achieve aligned user-preferable responses and train personalized Language Model (LM) assistants. Reinforcement Learning from Human Feedback (RLHF) (Stiennon et al., 2020; Ouyang et al., 2022; Christiano et al., 2017; Wang et al., 2023) demonstrates impressive results to achieve desired alignment, and utilizes a three-step approach of Supervised Fine-Tuning (SFT), reward model training, and policy optimization. However, its complexity presents several optimization and resource challenges in its multi-stage methodology. This has recently driven significant research into more resource efficient yet effective methods for human preference fine-tuning.

The Direct Preference Optimization (DPO) (Rafailov et al., 2024) algorithm proposes a simpler and more computationally lightweight alternative to aligning LLMs for user-preferred responses. DPO parametrizes the reward function instead of learning an explicit reward model and incorporates this into the Bradley-Terry ranking objective (Bradley & Terry, 1952). Although simpler, this also yields certain drawbacks: requiring a reference model to stabilize training incurs additional memory costs to host two models in order to train one model, additional computational costs due to its approximately $10\times$ smaller learning rate (HuggingFace Alignment Team, 2023), and a proven mismatch between the reward optimized in training and the log-likelihood optimized during inference (Meng et al., 2024) results in unstable reward margin gains. The authors of Chen et al. (2024) have also shown that ultimately models trained with DPO may exhibit random ranking accuracy even after time-extensive training. The more recent Simple Preference Optimization (SimPO) (Meng et al., 2024) algorithm employs an implicit reward formulation that directly aligns with the generation metric, thus eliminating the need for a reference model and yielding a more effective memory-efficient approach. However this also introduces additional hyperparameters, which the authors note are *crucial* to achieving reasonable performance. Other reference model-free methods, such as Odds

Ratio Preference Optimization (ORPO) (Hong et al., 2024), require additional hyperparameter tuning and extremely small learning rates on the order of $1 \times 10^{-9}$ or lower to achieve good performance, leading to generally the longest training time compared to other reference-free methods, yet potential stable reward margin gains.

In this paper, we introduce COALA: a novel, fast, and lightweight framework for preference fine-tuning LLMs efficiently on a single GPU. This timely line of research is motivated by recent advances that seek more efficient methods to probing the emergent behaviors of LLMs (Brown et al., 2024b), and making progress beyond simply saturating existing scaling laws. COALA leverages a *convex two-layer neural network* (cvxNN) on top of a pre-trained model for fast convergence with theoretical guarantees. Our algorithm mitigates the instability of offline preference fine-tuning through the introduction of a convex module for consistent reward margin increments. Unlike existing DPO style objectives, our novel COALA objective is solved using ADMM (Boyd et al., 2011) based algorithms for convex optimization in a scalable and near-hyperparameter-free manner. We implement our method in JAX (Bradbury et al., 2018) and use its Just-In-Time-Compilation (JIT) feature to lower COALA code into fast machine code for improved VRAM efficiency. As a result, COALA fine-tunes models such as LLaMA-8B and Mistral-7B on a single RTX-4090 GPU (NVIDIA Corporation, 2025) and performs competitively on benchmarks such as AlpacaEval2 (Li et al., 2023), ArenaHard (Li et al., 2024), and MT-Bench (Bai et al., 2024). Comprehensive experiments span five models $\times$ three datasets $\times$ four methods. Results presented include over **105** training runs, and TFLOPS metrics which validate COALA's substantial compute and power efficiency. Notably, COALA only requires $\approx 17.6\%$ of DPO's total TFLOPS in preference fine-tuning LLaMA-8B. Since alignment benchmarks are ultimately surrogate measures for alignment with real human preferences, we additionally conduct fair double-blind evaluation with **107 real human samples** to both assess the validity of benchmark scores with respect to actual human feedback, and to further establish the practical effectiveness of COALA in real world deployment.

**Contributions.** Our main contributions can be summarized as follows:

- We introduce COALA: a novel convex framework for efficient LLM preference alignment which achieves consistent improvements at significantly lower compute cost, thus enabling training on a single RTX-4090 GPU in a resource constrained environment study. Our theory-backed, compute-efficient algorithm is the first to successfully apply convex optimization to preference fine-tuning.

- We prove the COALA objective can be trained to global optimality in polynomial time using ADMM-based techniques. This theoretical guarantee ensures faster, more stable convergence and improved VRAM efficiency while mitigating reliance on extensive hyperparameter tuning. Results are validated across five models, three datasets, and by 107 real human evaluators.

- We develop EduFeedback: a custom conversational dataset of 26,621 conversations in an educational student-tutor setting. We introduce the novel **"Alternating Population Strategy"** to extract 65,606 preference chosen-rejected training samples. This method is significantly more efficient than tradition methods for creating chosen-rejected training triplets for preference fine-tuning, and avoids the need for sampling and re-ranking by a third-party model.

- We provide an open-source, modular JAX codebase to ensure easy reproducibility and application to new use-cases. The release includes support for Pytorch (Paszke et al., 2019) and multi-GPU settings for future research on scalability. We also provide open-source access to our EduFeedback dataset on Hugging Face (Hugging Face, 2023) for the machine learning community in support of ongoing research and experimentation.

## 2 RELATED WORK

Preference fine-tuning large language models (LLMs) can be classified as three distinct strategies. **(1)** Initial algorithms of zero-shot and few-shot in-context learning (Xian et al., 2017) rely on prompt engineering. This method does not require extensive compute, but is unable to tackle complex tasks and remains generally unreliable. **(2)** More sophisticated algorithms use reinforcement learning to align model output with user preferences. Successful algorithms in this class (such as RLHF

and RLAIF (Lee et al., 2023)) have been able to create conversational LLMs such as Google Gemini (Google DeepMind, 2023) and ChatGPT (OpenAI, 2023). However, despite their impressive performance, these methods are extremely complex, rely on expensive humans in the loop, and require significant computational resources in terms of both time and compute. **(3)** DPO-style methods leverage applications of the Bradley & Terry (1952) model to provide simple yet often performant learning algorithms that do not require explicit reward modeling. Yet these strategies depend critically on hyperparameter tuning to achieve good performance (Meng et al., 2024), require small learning rates typically on the order of $1 \times 10^{-9}$ or lower, thus leading to slow convergence and are VRAM expensive.

As LLMs become more widely adopted, there is also escalating awareness that scaling alone is increasingly unsustainable economically, environmentally, and in terms of diminishing returns. This has driven recent motivating work to make progress through avenues such as repeated sampling (Brown et al., 2024a), creative chain-of-thought reasoning (Wei et al., 2022), and various theoretically interpretable techniques (Singh et al., 2023). Therefore, single GPU methods are a timely and practical direction for preference alignment. Targeting a single device improves robustness (fewer distributed failure modes), latency (no cross-node synchronization), and data locality/security (privacy-sensitive deployment with on-premises options). This trend is reflected in recent systems that deliver applications in text, vision, and multimodal settings, such as FlexGen (Sheng et al., 2023), Cramming (Geiping & Goldstein, 2023), Inf-MLLM (Ning et al., 2024), Nimble (Kwon et al., 2020), and "Is one GPU enough?" (Tragakis et al., 2024). Naturally human preference fine-tuning is also currently positioned in this emerging paradigm, which seeks low-latency, resource-aware algorithms without sacrificing generative quality.

Bengio et al. (2005) have previously shown that it is possible to characterize the optimization problem for neural networks as a convex program. Pilanci & Ergen (2020) further developed exact convex reformulations for training a two-layer ReLU neural network. The reformulation uses semi-infinite duality theory to show that two-layer neural networks with ReLU activations and weight decay regularization may be re-expressed as a linear model with a group lasso penalty and polyhedral cone constraints. This yields both practical benefits in implementation and theoretical advantages in analyzing the optimization of the non-convex landscape of NNs. Bai et al. (2023) and Feng et al. (2024) have recently proposed cvxNN approaches based on the Alternating Direction Method of Multipliers (ADMM) (Boyd et al., 2011) to solve high-dimensional deep learning tasks. ADMM offers several advantages, such as its robustness against hyperparameter selection, linear decomposability for distributed optimization, and immunity to vanishing/exploding gradients. Its natural parallelization ability makes ADMM particularly suitable for optimization in deep learning, where scalability is crucial. Our novel COALA algorithm combines these features to preference-align LLMs faster with less memory and compute while maintaining competitive performance.

## 3 CONVEX NEURAL NETWORKS

This section provides background on convex neural networks (cvxNN), which is the basis for our introduction of the Convex Preference Optimization (COALA) algorithm. Our approach is motivated by defining a principled preference fine-tuning objective with strong convergence guarantees.

### 3.1 TWO-LAYER RELU NETWORKS

Given input $x \in \mathbb{R}^d$, the classic two-layer ReLU network is given by:

$$f(x) = \sum_{j=1}^{m} (\Theta_{1j}x)_+ \theta_{2j}, \tag{1}$$

where $\Theta_1 \in \mathbb{R}^{m \times d}$, $\theta_2 \in \mathbb{R}^m$ are weights of the first and last layer respectively, and $(\cdot)_+ = \max\{\cdot, 0\}$ is the ReLU activation function.

Given targets $y \in \mathbb{R}^n$, the network in equation 1 is trained by minimizing the following non-convex loss function:

$$\min_{\Theta_1, \theta_2} \ell\left(f_{\Theta_1, \theta_2}(X), y\right) + \frac{\beta}{2} \sum_{j=1}^{m} \left(||\Theta_{1j}||_2^2 + (\theta_{2j})^2\right), \tag{2}$$

where $\ell : \mathbb{R}^n \mapsto \mathbb{R}$ is the loss function, $X \in \mathbb{R}^{n \times d}$ is the data matrix, and $\beta \geq 0$ is the regularization strength. The non-convex nature of equation 2 makes its solution challenging, since the optimizer typically needs meticulous tuning of hyperparameters to ensure successful training. This requires many expensive iterations of running the optimizer across multiple hyperparameter configurations in a grid search to obtain good performance. This dramatically contrasts with the convex optimization framework, where algorithms have strong convergence guarantees and involve minimal hyperparameters. It is desirable to maintain the expressive capabilities of ReLU neural networks while still preserving the computational advantages of convex optimization.

### 3.2 CONVEX REFORMULATION

Pilanci & Ergen (2020) have shown equation 2 admits a convex reformulation, thus alleviating the inherent difficulties of the non-convex landscape in deep learning. When condition $m \geq m^*$ is satisfied for some $m \geq n+1$, the reformulation has the same optimal value as the original non-convex problem and no information is lost through reformulating equation 2.

The convex reformulation is based on enumerating the actions of all possible ReLU activation patterns on the data matrix $X$. These activation patterns act as separating hyperplanes, which essentially multiply the rows of $X$ by 0 or 1 and can be represented by diagonal matrices. For fixed $X$, the set of all possible ReLU activation patterns may be expressed as

$$\mathcal{D}_X = \left\{ D = \text{diag}\left(\mathbf{1}(Xv \geq 0)\right) : v \in \mathbb{R}^d \right\}.$$

The cardinality of $\mathcal{D}_X$ grows as $|\mathcal{D}_X| = \mathcal{O}\left(r(n/r)^r\right)$, where $r := \text{rank}(X)$ (Pilanci & Ergen, 2020). Given $D_i \in \mathcal{D}_X$, the set of vectors $v$ for which $(Xv)_+ = D_i X v$, is given by the following convex cone: $\mathcal{K}_i = \{v \in \mathbb{R}^d : (2D_i - I)Xv \geq 0\}$.

The exponential size of $\mathcal{D}_X$ make its complete enumeration impractical. Instead, we work with a subset based on sampling $P$ patterns from $\mathcal{D}_X$ for tractability:

$$\min_{(v_i, w_i)_{i=1}^P} \ell\left(\sum_{i=1}^P D_i X(v_i - w_i), y\right) + \beta \sum_{i=1}^P ||v_i||_2 + ||w_i||_2 \tag{3}$$

$$\text{s.t. } v_i, w_i \in \mathcal{K}_i \quad \forall i \in [P].$$

Although equation 3 is based on subsampling patterns in the convex reformulation, it can be shown under mild conditions that equation 3 still has the same optimal solution as equation 2 (Mishkin et al., 2022). The recent work of Kim & Pilanci (2024) also proves that the difference is negligible even when they are not equal. Therefore we can confidently work with the tractable convex program in equation 3.

In this paper we denote $\ell$ to be the mean-square error loss. The recent work of Bai et al. (2018) has shown that by adding slack variables, equation 3 can be written as:

$$\min_{v,s,u} ||Fu - y||_2^2 + \beta ||v||_{2,1} + \mathbb{I}_{\geq 0}(s) \quad \text{s.t. } u = v, \, Gu = s \tag{4}$$

The matrix $F \in \mathbb{R}^{n \times 2dP}$ is block-wise constructed by $D_i X$ terms.

## 4 COALA

In this section, we introduce our novel **C**onvex **O**ptimization for **A**lignment Preference **L**earning **A**lgorithm (COALA) and provide background on convex reformulated neural networks. Appendix B gives an explicit step-by-step through the COALA method.

### 4.1 CONVEX PREFERENCE OPTIMIZATION FRAMEWORK

In standard DPO, the goal is to obtain a good policy that is aligned with human preferences. The policy network $\pi_\theta$ is first initialized with the weights of a pre-trained network. It then aligns the policy model by solving the optimization problem

$$L_{\text{DPO}}(\pi_\theta; \pi_{\text{ref}}) = -\mathbb{E}\left[\log \sigma\left(\beta \log \frac{\pi_\theta(y_w|x)}{\pi_{\text{ref}}(y_w|x)} - \beta \log \frac{\pi_\theta(y_l|x)}{\pi_{\text{ref}}(y_l|x)}\right)\right] \tag{5}$$

The DPO objective in equation 5 is a large-scale non-convex optimization, which is challenging to solve. We first reformulate this learning problem using cvxNN, in order to admit more elegant optimization techniques. We adopt a modified Bradley-Terry model with an offset parameter $\gamma > 0$.

$$p(y_w \succ y_l | x) = \sigma(r(x, y_w) - r(x, y_l) - \gamma), \tag{6}$$

COALA then uses the un-normalized log-likelihood as its rewards function.

$$r(x, y) = \beta \log \pi(y|x). \tag{7}$$

Instead of taking $\pi$ in equation 7 to be a traditional neural network (NN) model, we replace it with a cvxNN. Specifically, we take a pre-trained model $f_{\theta_{\text{pre}}}(x)$ and stack a two-layer convex neural network $g^{\text{cvx}}_{\Theta_1, \theta_2}$ on top to serve as a binary preference classifier. $g^{\text{cvx}}_{\Theta_1, \theta_2}$ is then trained by solving the convex optimization problem in equation 4. Letting $\theta = (\theta_{\text{pre}}, \Theta_1, \theta_2)$, the resulting policy is then given by:

$$\pi^{\text{cvx}}_\theta(y|x) := \frac{1}{1 + \exp\left(-y g^{\text{cvx}}_{\Theta_1, \theta_2}\left(f_{\theta_{\text{pre}}}(x)\right)\right)}.$$

Instead of executing preference optimization with the weights of the entire model $g^{\text{cvx}}_{\Theta_1, \theta_2} \circ f_{\theta_{\text{pre}}}$, we freeze the weights of $f_{\theta_{\text{pre}}}$ and freeze $\Theta_1$ in $g^{\text{cvx}}_{\Theta_1, \theta_2}$. Inserting equation 7 into equation 6 with $\pi(y|x)$ replaced by $\pi^{\text{cvx}}_\theta(y|x)$, yields the COALA objective:

$$\min_{\theta_2} \ L_{\text{COALA}}(\pi^{\text{cvx}}_{\theta_2}) := \tag{8}$$

$$- \mathbb{E}_{(x, y_w, y_l) \sim \mathcal{D}} \left[\log \sigma\left(\beta \log \pi^{\text{cvx}}_{\theta_2}(y_w|x) - \beta \log \pi^{\text{cvx}}_{\theta_2}(y_l|x) - \gamma\right)\right],$$

Notably, equation 8 is reference-free in comparison to equation 5 yet does not incur additional scaling hyperparameters for stability. This is supported in recent work such as SimPO (Meng et al., 2024) and ORPO (Mirhoseini et al., 2024), where we find the reference model unnecessary.

In addition to improved compute and memory efficiency by removing the reference model, the advantage of solving equation 8 over equation 5 also gives higher computational tractability. The following proposition shows equation 8 is convex.

**Proposition 1** (COALA Loss is Convex). *The optimization problem in equation 8 may be written as*

$$\min_{\theta_2} \ \mathbb{E}\left[\log\left(1 + \exp\left(-\beta y_w \theta_2^T (\Theta_1 f_{\theta_{\text{pre}}}(x))_+ + \gamma\right)\right)\right]. \tag{9}$$

*This objective is convex as equation 9 is a logistic regression problem in $\theta_2$.*

The proof of proposition 1 is provided in section A. Proposition 1 shows that $L_{\text{COALA}}$ is convex. Thus, we can solve it to global optimality in polynomial time using more efficient gradient-based optimizers. Since we only have to train the weights of the final layer of the convex model, COALA requires significantly less computation than existing methods and provides the second major reason why COALA can quickly train on one GPU.

## 4.2 COALA ALGORITHM

---

**Algorithm 1** Convex Preference Optimization (COALA)

---

**Require:** Dataset $(x, y_w, y_l)$, Pre-trained model $f_{\theta_{\text{pre}}}(x)$, offset parameter $\gamma$, penalty parameter $\rho > 0$
    **Phase I: Train the policy network**
    Train $\pi^{\text{cvx}}$ to obtain $(\Theta_1, \theta_2)$ by solving equation 4 using CRONOS($\rho$) (Algorithm 2).
    **Phase II: Finetuning**
    Freeze weights of the first layer $\Theta_1$.
    Finetune weights of second layer $\theta_2$ by solving the convex minimization problem equation 9:
        $\min_{\theta_2} L_{\text{COALA}}(\pi^{\text{cvx}}_{\theta_2})$
**Ensure:** $(\Theta_1, \theta_2)$

---

We formally present the pseudocode for COALA in algorithm 1. Stage one of COALA first trains a cvxNN on top of a standard pre-trained model. In stage two, the final policy model is obtained by solving a convex logistic regression problem in a fine-tuning step.

The key to maximizing COALA's efficiency in this framework is using the recently introduced CRONOS algorithm (Feng et al., 2024) to train $\pi_\theta^{\text{cvx}}$, which we now discuss in detail.

### 4.2.1 Efficiently Training the Convex Policy Network via CRONOS

---

**Algorithm 2** CRONOS with Preconditioned Conjugate Gradient (PCG)

---

**Require:** penalty parameter $\rho$
  **repeat**
    $\boldsymbol{u}^{k+1} = \text{argmin}_{\boldsymbol{u}} \left\{ \frac{1}{2}\|F\boldsymbol{u} - y\|^2 + \frac{\rho}{2}\|\boldsymbol{u} - \boldsymbol{v}^k + \lambda^k\|_2^2 + \frac{\rho}{2}\|G\boldsymbol{u} - \boldsymbol{s}^k + \nu^k\|^2 \right\}$    $\triangleright$ Use PCG
    $\begin{bmatrix} \boldsymbol{v}^{k+1} \\ \boldsymbol{s}^{k+1} \end{bmatrix} = \text{argmin}_{\boldsymbol{v},\boldsymbol{s}} \left\{ \beta\|\boldsymbol{v}\|_{2,1} + \mathbf{1}(\boldsymbol{s} \geq 0) + \frac{\rho}{2}\|\boldsymbol{u}^{k+1} - \boldsymbol{v} + \lambda^k\|^2 \right\}$    $\triangleright$ Primal update
    $\lambda^{k+1} \leftarrow \lambda^k + \frac{\gamma_\alpha}{\rho}(\boldsymbol{u}^{k+1} - \boldsymbol{v}^{k+1})$    $\triangleright$ Dual $\lambda$ update
    $\nu^{k+1} \leftarrow \nu^k + \frac{\gamma_\alpha}{\rho}(G\boldsymbol{u}^{k+1} - \boldsymbol{s}^{k+1})$    $\triangleright$ Dual $\nu$ update
  **until** convergence

---

CRONOS (Feng et al., 2024) is a specialized version of the Alternating Direction Method Multiplers (ADMM) Boyd et al. (2011), for solving the optimization problem in equation 4. The choice of ADMM for solving equation 4 in Feng et al. (2024) is predicated on three key properties: **(1)** It possesses a robust convergence guarantee as shown in Section 4.3, **(2)** it lifts memory constraints in LLM classification problems, and **(3)** it is highly optimized for GPU acceleration in JAX.

Algorithm 2 formally presents using CRONOS for solving eq. (4). Two subproblems must first be solved efficiently, then the algorithm only requires vector addition and one matrix-vector product. The structure of these subproblems makes CRONOS readily compatible with hardware accelerators and JAX.

### 4.3 COALA Convergence Guarantees

Since COALA interprets the preference alignment task as a convex optimization problem application, it also immediately inherits the rich convergence theory associated with convex algorithms. In particular, we show that this leads to convergence guarantees for each aspect of the COALA pipeline and robustness to hyperparameters tuning. The following result from Feng et al. (2024) shows CRONOS converges ergodically at an $\mathcal{O}(1/k)$-rate.

**Theorem 1** (Convergence of ADMM for equation 4). *Let $\{\delta_k\}_{k\geq 1}$ be some summable sequence. Run Algorithm 2 and suppose at each iteration the computed $u^{k+1}$ satisfies:*

$$\left\| u^{k+1} - \text{argmin}_u \left\{ \frac{1}{2}\|Fu - y\|^2 + \frac{\rho}{2}\|u - v^k + \lambda^k\|_2^2 + \frac{\rho}{2}\|Gu - s^k + \nu^k\|^2 \right\} \right\| \leq \delta_k.$$

*Therefore after $K$ iterations, the output of CRONOS algorithm 2 satisfies:*

$$\|F\bar{u}^K - y\|^2 + \beta\|\bar{v}^K\|_{2,1} + \mathbf{1}(\bar{s}^K \geq 0) - p^\star = \mathcal{O}(1/K),$$

$$\left\| \begin{bmatrix} I_{2dP} \\ G \end{bmatrix} \bar{u}^K - \begin{bmatrix} \bar{v}^K \\ \bar{s}^K \end{bmatrix} \right\| = \mathcal{O}(1/K).$$

This guarantee holds for any $\rho > 0$ and when the $u$-subproblem is solved inexactly. Consequently, CRONOS' convergence is robust, making it an ideal subroutine for training the cvxNN in COALA. This ensures efficient and successful training of the policy network.

The COALA fine-tuning step also has strong guarantees, since the minimization objective in equation 9 is smooth, convex, and has a **Lipschitz continuous gradient**. The latter property follows as the logistic loss has a Lipschitz continuous gradient. Thus, if we apply Accelerated Gradient Descent (AGD) (Nesterov, 1983; d'Aspremont et al., 2021), which has the worst-case optimal convergence rate, to solve equation 9, we obtain the following result.

**Theorem 2** (Efficient minimization of COALA loss eq. (9)). *Suppose we run AGD to solve equation 9. Then after $k$ iterations, the output $\theta_2^k$ satisfies:*

$$L_{\text{COALA}}(\pi_{\theta_2^k}^{\text{cvx}}) - \min_{\theta_2} L_{\text{COALA}}(\pi_{\theta_2}^{\text{cvx}}) = \mathcal{O}(1/k^2).$$

Theorem 2 shows we can train the COALA loss to global optimality in polynomial time. This contrasts greatly with DPO, which is non-convex and lacks convergence guarantees. In addition, optimizer methods in methods which are DPO-based is typically non-trivial and requires a learning rate 10 times smaller than a respective SFT learning rate (Meng et al., 2024).

## 5 EXPERIMENTS

We experiment with five models: DistilGPT (Sanh et al., 2019), GPT-2 (Radford et al., 2019), Mistral-7B (Jiang et al., 2023a), Dolphin2.6-7B (Cognitive Computations, 2024), and LLaMA-8B (Touvron et al., 2023) on three datasets against four preference alignment algorithms. In order to ensure fair comparison, we run each experiment configuration on a single A100 GPU with 40GB VRAM, with the exception of COALA experiments which run on a single RTX-4090 GPU with 24GB VRAM. This ablation is necessary, since methods (Appendix E.1) such as DPO and ORPO require more memory on the same datasets. All numerical results are presented after averaging over three runs in each setting.

### 5.1 DATASETS

This section introduces the three main datasets used for preference fine-tuning and SFT in our experiments. Details on preference dataset extraction specifics are shown in Appendix E.2.

- **EduFeedback.** This is our synthetically generated conversational dataset inspired by Ultra-Chat (Ding et al., 2023), but clearly constrained within an educational setting. EduFeedback contains 26,621 conversations generated with GPT-4o (OpenAI, 2024). We vary $t = 0.2 - 0.9$, utterances range from 4-8 alternating turns between two agents. We randomly vary *mood* of agents to simulate realistic human conversation, and encompass eleven diverse topics in fields of study such as science and philosophy. We introduce the novel *Alternating Population Strategy* to extract 65,606 preference training samples. More discussion and examples are given in Appendix E.3.
- **UltraFeedback, (Cui et al., 2023).** The original Ultrafeedback dataset comprises 64,000 training samples each containing four model-generated completions from a variety of open-source models. GPT-4 (OpenAI, 2023) was used to assign "preference scores" to each completion. This dataset was selected to be consistent with prior work, and serves as a standard equivalent benchmark.
- **IMDb, (Maas et al., 2011).** This IMDb positive-negative sentiment analysis dataset is selected to be consistent with prior work Rafailov et al. (2024). In each instance the "prompt" is either negative or positive, and generation quality is graded on the adherence and cogency of the of the output with regards to sentiment. We downsample this dataset to 11,000 training samples.

### 5.2 MODEL ARCHITECTURE DETAILS

**Baseline Models.** The five baseline models are SFT fine-tuned for one epoch on each of the three datasets before preference extraction. This creates a total of fifteen baseline models. Each method is then evaluated on the three preference extracted datasets, in "prompt", "chosen", and "rejected" triplet format. We experiment both with initiating preference training from a respective SFT baseline trained model, and directly starting a pre-trained checkpoint that has not been SFT trained to be in distribution with the task. In all cases we aim to ensure fair comparisons between runs: by ensuring SFT trained case models see the same amount of data and averaging three runs per setting in all results.

**Preference Fine-tuned Models.** Preference fine-tuning utilizes the EduFeedback-Alternate, UltraFeedback-Binarized, and IMDb datasets. We experiment with five model architectures for preference alignment: DistilGPT, GPT-2, Mistral-7B, Dolphin2.6-7B, and LLaMA-8B. The Dolphin2.6-7B

model is selected for its existing *instruction fine-tuned* checkpoint. Appendix E.1 provides more details on each of the bench-marked methods, and Table 8 summarizes crucial hyperparameters and objective functions for our main methods of interest.

# 6 MAIN RESULTS AND DISCUSSION

In this section, we present main experimental results showing the stable and monotonically increasing reward margins of COALA. We also present comprehensive ablation studies on the effectiveness of initializing training from an SFT fine-tuned base, and quantitatively assess the effectiveness of preference alignment methods on AlpacaEval2. Comprehensive performance plots, MT-Bench scores, ArenaHard, TFLOPS measurements and further details are presented in Appendix C and D.2. We critically evaluate with 107 real human participants to validate the effectiveness of COALA, since auto-run metrics are ultimately surrogates for modeling real human preference. Details on our extensive human study is presented in Appendix G and a summary of numerical results is presented in Table 2 .

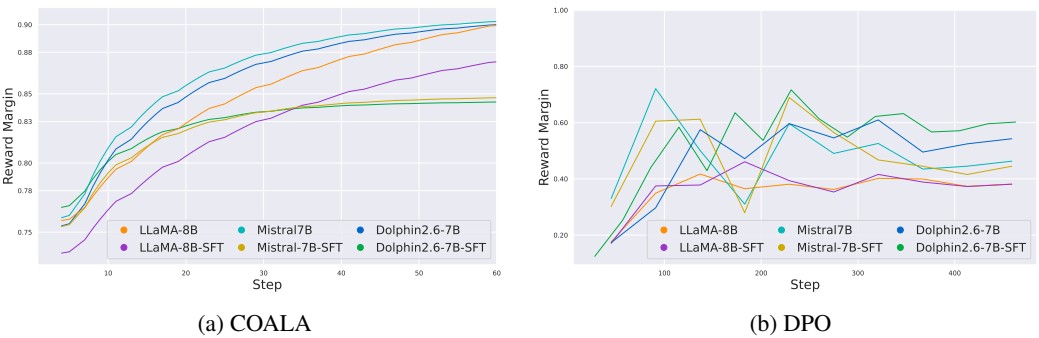

(a) COALA             (b) DPO

Figure 1: Our novel COALA algorithm shows stable reward margin gains across all models, due to its theoretically grounded foundation which alleviates fine-tuning reliance on hyperparameter tuning and general heuristics.

| Method | LC WR % | | | WR % | | | Avg Length | | |
|---|---|---|---|---|---|---|---|---|---|
| | Edu | IMDb | Ultra | Edu | IMDb | Ultra | Edu | IMDb | Ultra |
| *Mistral-7B Model* | | | | | | | | | |
| COALA | 24.61 | 24.88 | 20.84 | 23.82 | 23.11 | 20.91 | 592 | 418 | 459 |
| ORPO | 17.01 | 17.58 | 16.04 | 14.67 | 15.62 | 12.28 | 561 | 368 | 350 |
| DPO | 24.19 | 24.30 | 17.68 | 22.45 | 22.74 | 15.56 | 492 | 502 | 453 |
| SFT | 6.80 | 8.42 | 6.18 | 10.30 | 1.77 | 6.11 | 515 | 428 | 463 |
| *Dolphin-7B Model* | | | | | | | | | |
| COALA | 40.81 | 39.72 | 31.58 | 39.05 | 38.46 | 30.21 | 439 | 454 | 445 |
| ORPO | 25.06 | 24.90 | 22.94 | 23.59 | 22.92 | 25.93 | 526 | 448 | 452 |
| DPO | 34.73 | 33.86 | 26.41 | 32.46 | 31.58 | 24.86 | 494 | 511 | 476 |
| SFT | 17.36 | 16.21 | 14.88 | 15.30 | 15.47 | 12.76 | 404 | 423 | 459 |
| *Llama-8B Model* | | | | | | | | | |
| COALA | 40.90 | 27.64 | 20.64 | 38.20 | 25.68 | 18.32 | 562 | 415 | 552 |
| ORPO | 23.87 | 12.10 | 12.91 | 20.58 | 12.05 | 10.95 | 599 | 610 | 354 |
| DPO | 40.68 | 21.79 | 18.89 | 38.53 | 20.18 | 15.81 | 539 | 449 | 503 |
| SFT | 10.92 | 8.16 | 7.41 | 10.75 | 8.11 | 5.62 | 384 | 435 | 546 |

Table 1: AlpacaEval2 Metrics by Alignment Method for Three Models on Three Datasets

## 6.1 STABLE INCREASE IN REWARDS

One of the most prominent advantages of COALA is its stable increase in reward margin gains, as seen in Figure 1. Preference fine-tuning of LLMs is well known to be noisy and unstable in training,

especially in offline settings. While the additional reference model in traditional DPO seeks to mitigate this, it also becomes immensely VRAM expensive. More details are presented in Appendix F, and Table 3 summarizes TFLOPS measurements per method. In contrast, COALA effectively stabilizes the preference alignment task by re-framing this as a convex optimization problem with principled convergence guarantees. Since rewards steadily increase over time, this eliminates the need for an exponentially smaller learning rate, such as in DPO and ORPO. The integration of CRONOS to solve the cvxNN training task further reduces the fine-tuning time by an order of magnitude (see Appendix F) and allows successful fine-tuning of LLMs even on one GPU by lifting certain dimension constraints.

## 6.2 EFFECT OF SFT BASELINE AND DATASETS

Training an SFT baseline adds additional complexity and cost to preference fine-tuning, but in most cases is beneficial. Table 6 shows that the SFT training alone is often able to achieve significant performance gains in terms of preference alignment. This is especially true in larger more expressive models such as LLaMA-8B. In contrast, offline DPO alone often does not surpass the performance of an SFT trained LLM that is sufficiently pre-trained. ORPO delivers the slowest training for one epoch, but is able to preference align over time. COALA is able to perform competitively and consistently in significantly shorter time, memory, and compute (see TFLOPS measurements in Appendix D). This agrees with our expectations, since the large amount of pre-training experienced in LLMs is inherently expressive, and principled preference alignment is able to guide the LLMs without exhaustive compute-extensive training.

## 6.3 VALIDATION FROM REAL HUMAN FEEDBACK

In order to validate our objective to fine-tune LLMs to human preference, we conduct double-blind experimentation with real human participants. Appendix G details our 107-sample real human study with associated sample questions and percentage win rates. Appendix G.3 presents the consent form to participate in the study for ethical considerations. Fifty persons are from a deep learning class, and fifty seven persons are from a commercial technology

|      | COALA  | ORPO  | DPO   | SFT   |
|------|--------|-------|-------|-------|
| Edu  | **39.1%** | 15.5% | 28.8% | 16.6% |
| IMDb | **42.7%** | 20.1% | 24.8% | 12.4% |

Table 2: 107 Real Human Feedback Win Rates per Method and Dataset

sector of the deep learning industry. COALA exhibits the highest real human preference on average across all questions, as seen in Table 2. This experiment is in the same spirit as the 24-sample human study conducted in the seminal DPO paper, and also highlights the existing gap between automated evaluation metrics versus real human preferences in sensitive alignment tasks.

## 6.4 EXPRESSIVENESS TRADEOFF

Freezing the base model parameters can constrain the policy from learning deep semantic shifts (e.g., creative writing style changes). The main advantage of COALA is its stable reward margin gains, sustainable yet effective compute efficiency, and bridging the gap between theoretically founded derivatives with validated real-world human preference gains. Therefore this method targets a critical and widespread class of alignment problems in concise and objective preference ranking (e.g., correctness, helpfulness, and safety) rather than subjective style transfer. This design choice is also motivated by recent literature (Cava & Tagarelli, 2025; Swamy et al., 2025), which suggest that full parameter modification is often unnecessary for effective preference alignment. Other authors (Cao et al., 2024; Rimsky et al., 2024; Kowsher et al., 2025) have shown that lightweight steering vectors or residual modifications can achieve comparable gains with significantly less compute. COALA effectively offers a principled, stable, and compute-efficient alternative for these objective-focused tasks, positioning it as an ideal solution for resource-constrained or on-device settings such as personalized LLM assistants or model predictive control navigational scenarios.

## 7 CONCLUSION

We introduce COALA, a convex optimization-based framework for preference learning that is stable, efficient, and reference-free. The key insight of COALA is in recognizing that preference learning

can be framed as **convex optimization** based task. The challenge is capturing the expressiveness of the rich features in the underlying base model in a principled and efficient method. By using a convex neural network (cvxNN) to provide a stabilized reward signal, COALA eliminates the need for a frozen reference model and reduces computational overhead. Instead of optimizing over input text, it operates on preference features derived from the foundational LLM, thus preserving expressiveness while lowering input dimensionality. Our JAX-based implementation leverages CRONOS and ADMM for efficient parallelization with fast iteration on a single GPU. COALA demonstrates strong performance across three datasets, shows robustness to hyperparameter tuning, and is validated by 107 samples of real human feedback. By lowering the resource barrier for alignment, COALA opens preference fine-tuning to broader educational and research use. Future work will explore a broader set of multilingual preference datasets and include more detailed studies in data modalities. A deeper understanding of how large language models internalize preference signals during alignment can yield more efficient fine-tuning with wider applications. Additional discussions on limitations, future work, and expressiveness tradeoffs are presented in Appendix H.

## ACKNOWLEDGMENTS

Omitted during submission for reviews for double-blind anonymity.

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

## A  PROOF THAT COALA LOSS IS CONVEX (PROPOSITION 1)

*Proof.* Recall the COALA objective is given by

$$\min_{\theta_2} -\mathbb{E}_{(x,y_w,y_l)\sim\mathcal{D}} \left[ \log \sigma \left( \beta \log \frac{\pi_{\theta_2}^{\text{cvx}}(y_w|x)}{\pi_{\theta_2}^{\text{cvx}}(y_l|x)} - \gamma \right) \right].$$

For COALA, we have that

$$\pi_{\theta_2}^{\text{cvx}}(y_w|x) = \frac{1}{1 + \exp\left(-y_w(\theta_2^T \tilde{x})\right)},$$

where $\tilde{x} = (\Theta_1 f_{\theta_{\text{pre}}}(x))_+$. Using this expression for $\pi_{\theta_2}^{\text{cvx}}(y_w|x)$, we find

$$\begin{aligned}
\beta \log \frac{\pi_{\theta_2}^{\text{cvx}}(y_w \mid x)}{\pi_{\theta_2}^{\text{cvx}}(y_l \mid x)} &= \beta \log \left( \frac{\pi_{\theta_2}^{\text{cvx}}(y_w|x)}{1 - \pi_{\theta_2}^{\text{cvx}}(y_w|x)} \right) \\
&= \beta \log \left( \exp\left( y_w \theta_2^T \tilde{x} \right) \right) \\
&= \beta y_w \left( \theta_2^T \tilde{x} \right)
\end{aligned}$$

From this last display and the definition of the sigmoid function, we immediately deduce

$$\log \sigma \left( \beta \log \frac{\pi_{\theta_2}^{\text{cvx}}(y_w|x)}{\pi_{\theta_2}^{\text{cvx}}(y_l|x)} - \gamma \right) = - \log \left( 1 + \exp \left( -\beta y_w \left( \theta_2^T \tilde{x} \right) + \gamma \right) \right).$$

Thus, using the last display and the definition of $\tilde{x}$, the Convex DPO objective may be rewritten as,

$$\min_{\theta_2} \mathbb{E}_{(x,y_w,y_l)\sim\mathcal{D}} \left[ \log \left( 1 + \exp \left( -\beta y_w \theta_2^T (\Theta_1 f_{\theta_{\text{pre}}}(x))_+ + \gamma \right) \right) \right].$$

The Convex DPO objective is a logistic regression problem in $\theta_2$ and thus is convex. □

## B  COALA STEP-BY-STEP

We explicitly step through the COALA method for enhanced clarity. In the following example, we assume the base model being finetuned is LLaMA-8B and state respective input dimensions.

**Phase I: Train the Convex Policy Network**

- **Feature Extraction:** Take a pre-trained language model, denoted as $f_{\theta_{\text{pre}}}(x)$ (e.g., LLaMA-8B) and extract the last-layer hidden states (embeddings) for the input data $x$. For the LLaMA-8B model, these extracted features have a dimension of $d = 4096$.
- **Convex Reformulation (cvxNN):** Instead of a standard neural network head, COALA stacks a convex neural network (cvxNN) on top of these $d = 4096$ extracted features. This network, denoted as $g_{\Theta_1,\theta_2}^{\text{cvx}}$, serves as a binary preference classifier.
- **Solving with CRONOS:** We train this cvxNN by solving the convex optimization problem (Equation 4) using the CRONOS algorithm. CRONOS uses the Alternating Direction Method of Multipliers (ADMM) to efficiently solve this high-dimensional problem on a GPU. The output of this phase are the proven optimal weights for the convex layers: $(\Theta_1, \theta_2)$.

**Phase II: Preference Fine-Tuning**

- **Freeze Components:** We freeze the pre-trained base model weights ($f_{\theta_{\text{pre}}}$) and the first layer weights of the convex network ($\Theta_1$) obtained in Phase I.

- **Define the COALA Objective:** COALA uses a modified Bradley-Terry model for prefer­ence learning. The reward function is defined as the un-normalized log-likelihood of the convex policy: $r(x, y) = \beta \log \pi(y|x)$. The specific objective function ($L_{\text{COALA}}$) is derived by substituting the convex policy into the logistic loss formula. Crucially, this objective function is convex with respect to the weights ($\theta_2$).

- **Convex Minimization:** We fine-tune *only* the second layer weights ($\theta_2$) by minimizing the $L_{\text{COALA}}$ objective. Since the problem is convex, it can be solved to global optimality.

- **Final Policy:** The resulting aligned policy is a composition of the frozen pre-trained model and the trained convex head:

$$\pi_\theta^{\text{cvx}}(y|x) = \frac{1}{1 + \exp\left(-y g_{\Theta_1, \theta_2}^{\text{cvx}}(f_{\theta_{\text{pre}}}(x))\right)}$$

## C  ADDITIONAL EMPIRICAL RESULTS

### C.1  EFFICIENCY OF COALA

Table 3 displays the average TFLOPS usage across three runs on the EduFeedback dataset. COALA exhibits significantly lower TFLOPS on the same dataset, largely due to its extremely fast optimization techiniques. ORPO displayed the longest training times, but was the most robust to SFT versus non-SFT training baselines. Given enough time, ORPO seemed to product more ravorable results against DPO in terms of human feedback in Appendix G

| Model | COALA | DPO | ORPO | SFT |
|---|---|---|---|---|
| distilgpt2 | **152.56** | 537.12 | 643.33 | 271 |
| gpt2 | **379.89** | 1087.45 | 1305.27 | 522 |
| mistral-7B | **1580.45** | 9284.71 | 11241.89 | 2492 |
| llama-8B | **1805.39** | 10253.37 | 12352.98 | 2851 |
| dolphin-7B | **1794.66** | 10091.25 | 12116.50 | 2804 |

Table 3: TFLOPS measurements for COALA (from logs), DPO, ORPO (measured via NVSIGHT), and SFT (from logs) on the EduFeedback dataset.

## C.2 ADDITIONAL PLOTS

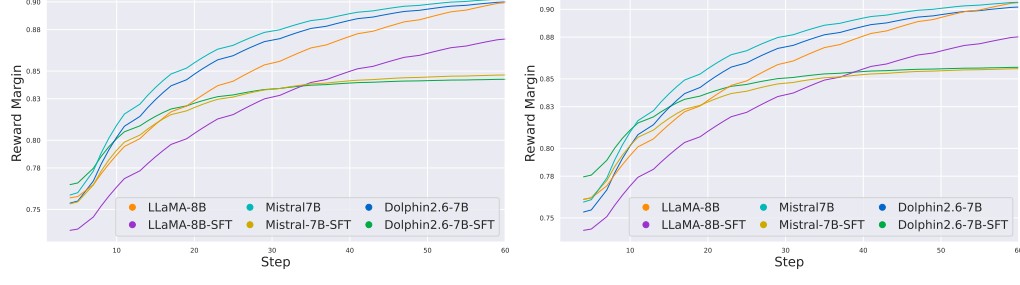

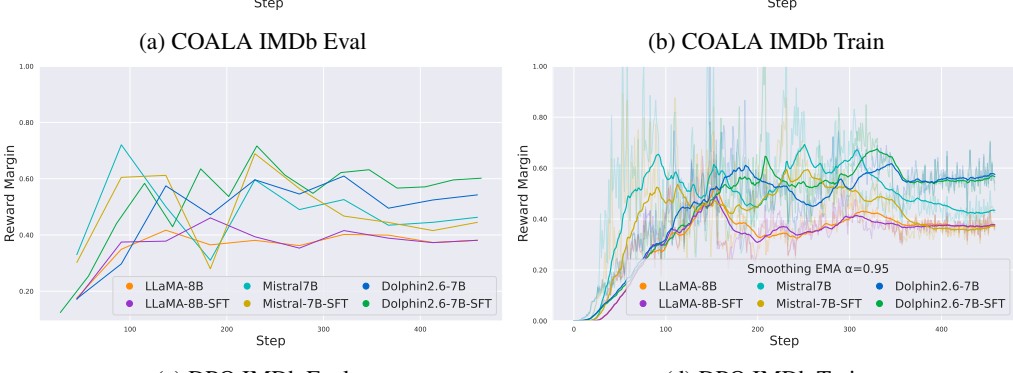

Figure 2: COALA and DPO mean reward margins for runs on the IMDb Dataset. The DPO Train dataset variant displays Time Weighted Exponential Moving Average (EMA) smoothing with the specified $\alpha$ parameter.

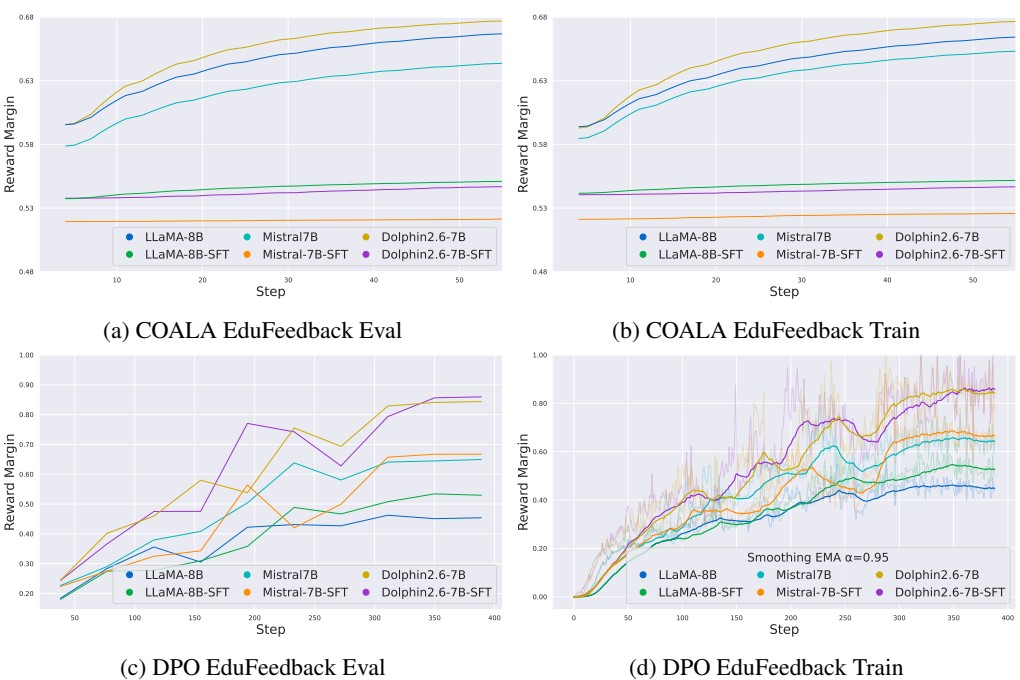

Figure 3: COALA and DPO mean reward margins for runs on the EduFeedback Dataset. The DPO Train dataset variant displays Time Weighted Exponential Moving Average (EMA) smoothing with the specified $\alpha$ parameter.

## C.3 GENERATIVE EXAMPLES PER METHOD

Table 4 provides examples of generated output to given prompts for different preference alignment methods. In each case we constrain generated tokens to be the same, in order for fair assessment. We also provide the MT-Bench comments by GPT-4 as judge in order ot be consistent with previous work (Rafailov et al., 2024).

---

**Prompt**
"prompt": "I have an exam coming up, what does glucose do in the human body?"

**COALA**
"response": "What does glucose do? I will explain its role in cell metabolism. Glucose is a simple sugar that serves as a primary energy source for cells. In cell metabolism, glucose is broken down through a series of chemical reactions known as glycolysis in the cytoplasm. This process produces a small amount of ATP (adenosine triphosphate), which is the primary energy currency of the cell. Additionally, it generates NADH (reduced nicotin"

**DPO**
"response": "Glucose is a simple sugar and a vital compound in the human body. Glucose is primarily responsible for doing the following:1. Energy Source: Glucose is the main source of energy for cells in the body. It is used by the body's cells, particularly the brain, muscles, and organs, to produce energy through a process called cellular respiration.2. Fuel for the Brain: Glucose is the main fuel for the brain, providing the energy needed for cognitive functions, memory, and mental activity.3. Storage of Energy: Glucose is stored in the body as glycogen in the liver and muscles. It serves"

**ORPO**
"response": "Glucose is a type of sugar that serves as a primary source of energy for cells in your body. It's particularly important for the brain, which relies heavily on glucose for energy. Glucose is absorbed from the food you eat, primarily from carbohydrates, and is transported in the blood to the cells. Insulin, a hormone produced by the pancreas, helps cells absorb glucose from the blood to use as energy. Glucose also plays a role in the production of ATP, the energy currency of the cell, which is used for various cellular activities. Maintaining stable levels of glucose in the blood is crucial for overall health and energy levels"

**SFT**
"response": "Glucose, a simple sugar, plays a crucial role in the body. It serves as a primary source of energy for the cells, especially for the brain and muscles. Glucose is absorbed from the food we eat and is transported to cells through the bloodstream. Within cells, glucose can be converted into ATP (adenosine triphosphate), which is the primary energy currency of the cell, enabling various cellular processes. Additionally, glucose is essential for synthesizing other important molecules, such as fatty acids and amino acids, which are crucial for growth, repair, and maintenance of tissues in the body.2: How do cells actually use"

**Judgment-GPT-4**
Response A **[COALA]** for an exam-prep scenario, provides the most biochemically grounded explanation, naming key steps and molecules (glycolysis, ATP, NADH) that demonstrate conceptual understanding — crucial for scoring well on technical test questions. Response B and C are well-organized and factually correct, but reads more like a textbook summary and doesn't include cellular details.

---

**Prompt**
"prompt": "After watching the film, I felt"

**COALA**
"response": "After watching the film, I felt that I could really relate to the character of Rachel. Like Rachel, I am a young woman who has always been very ambitious and has always had very high standards for herself. In the film, Rachel's ambition leads to her becoming a successful businesswoman, but it also leads to her becoming very obsessed with her work, which makes her neglect her relationships with others. This is a very common problem for ambitious young people, and it is something that I have struggled with myself. The film shows how Rachel's"

**DPO**

"response": "It's wonderful that I felt moved by the film! T̈he Lion Kingïs a truly beautiful and emotional story that touches upon themes of love, loss, and the importance of family. I can understand why you felt the following:1. **Powerful and emotional:** The film has an incredible ability to make you feel a wide range of emotions, from joy to sorrow, laughter to tears. It's a rollercoaster of emotions that really draws you in and keeps you engaged throughout. 2. **Amazing animation:** The animation in T̈he Lion Kingïs simply breathtaking. The attention to detail and the lifelike quality of"

**ORPO**

"response": "It's wonderful to hear that you enjoyed the film! I can understand why you felt that way. The film was beautifully crafted, with its captivating storyline, stunning visuals, and talented cast. It's no wonder that it left you feeling emotional and moved. The characters were so well-developed and the performances were outstanding. The film's themes of love, loss, and redemption were skillfully woven together, creating a powerful and impactful narrative. I'm glad that you were able to connect with the characters and the story on such a deep level. The film's exploration of the human experience and the complexities of relationships resonated with you, leaving"

**SFT**

"response": "It is important to note that you enjoyed a particular scene in the movie. Can you tell me more about it and why it was so memorable for you?"

**Judgment-GPT-4**

Response A **[COALA]** delivers a natural and insightful monologue continuation that reads like a real person sharing their post-film feelings. While Response B includes strange characters and symbols.

Table 4: Examples of generated output on EduFeedback data (first question) and IMDb positive sentiment generation (second question) for methods executed on LLaMA-8B base without SFT fine-tuning.

# D ABLATION STUDIES

## D.1 LEVERAGING CRONOS VERSUS VANILLA ADAMW

COALA's Stage One optimization must be as accurate, robust, and fast as possible in order to generate preferred responses. This is since without high accuracy and expressiveness in capturing preference features, COALA will be unable to effectively generate in Stage Two. If during Stage One optimization multiple seed sweeps, tuning of learning rates and other hyperparameters exist, then this negates the efficiency of COALA in preference alignment. Additionally in order to fit all COALA runs onto one RTX-4090 GPU, memory efficiency is critical and Stage Once optimization cannot be constrained by high-dimensional features. Therefore we utilize the CRONOS algorithm for its robustness, speed, and memory efficiency. CRONOS also eliminates the need for hyperparameter grid search associated with traditional optimizers such as AdamW. This is significant since poor hyperparameter selection in these optimizers might lead to non-convergence (Feng et al., 2024) and sub-optimal classification. Table 5 demonstrates the classification accuracy of CRONOS versus vanilla AdamW.

## D.2 ADDITIONAL NUMERICAL RESULTS

This section provides comprehensive metrics for our four main methods on five models across three datasets. Notably, **COALA consistently out-performs competing methods**. Table 6 reports MT-Bench scores, Table 7 reports ArenaHard scores, and Table 1 in the main work presents AlpacaEval2 scores. AlpacaEval2 is a popular benchmark selected for its success in assessing coherence and contextual understanding in conversational settings. We also select MT-Bench and ArenaHard for comprehensive evaluation and in order to be consistent with existing work (Meng et al., 2024). We average the scores of of each setting across twenty questions per judge, and validate the results of LLM judges with 107 human judgments. For preference fine-tuned models presented, we initiate

| Base Model | Ultra | | Edu | | IMDb | |
|---|---|---|---|---|---|---|
| | CRONOS | AdamW | CRONOS | AdamW | CRONOS | AdamW |
| distillGPT-82M | 53.50% | 53.80% | 60.60% | 56.18% | 83.05% | 82.62% |
| GPT2-774M | 53.83% | 52.93% | 61.15% | 59.80% | 78.84% | 78.78% |
| Mistral-7B | 57.63% | 52.95% | **69.83**% | 62.02% | **91.76**% | 83.52% |
| Dolphin-7B | 62.27% | 56.33% | 72.29% | 69.43% | **93.72**% | 83.40% |
| Llama-8B | **57.68**% | 51.20% | 71.01% | 68.55% | **90.43**% | 86.60% |

Table 5: Comparison of COALA Stage One classification accuracy of extracted features from SFT fine-tuned base models using CRONOS versus AdamW.

from a pre-trained SFT base model for stronger performance. We restrain the number of generated tokens to the same length in all experiments for fair comparison, since Celikyilmaz et al. (2020) has shown that often human users will prefer simply the longer generated output. In addition to evaluation in length-controlled win rate, we use the suggested `alpaca_eval_gpt4_turbo_fn` annontator, and default to GPT-4 as judge.

| Base Model | SFT | | | DPO | | | ORPO | | | COALA | | |
|---|---|---|---|---|---|---|---|---|---|---|---|---|
| | Ultra | Edu | IMDb | Ultra | Edu | IMDb | Ultra | Edu | IMDb | Ultra | Edu | IMDb |
| distilGPT | 1.6 | 1.6 | 1.2 | 1.2 | 1.0 | 1.6 | 1.4 | 1.0 | 1.2 | 1.0 | 1.7 | 1.2 |
| GPT-2 | 1.7 | 1.7 | 1.4 | 1.4 | 1.3 | 1.0 | 1.1 | 1.6 | 1.1 | 1.2 | 1.6 | 1.1 |
| Mistral-7B | 3.6 | 8.1 | 2.3 | 1.9 | 6.1 | 2.8 | 1.2 | 6.8 | 1.7 | 3.5 | 6.9 | 2.9 |
| Dolphin-7B | 2.7 | 8.3 | 3.2 | 4.4 | 8.3 | 5.1 | 5.4 | 7.9 | 5.5 | 5.5 | 8.1 | 5.6 |
| LLaMA-8B | 5.0 | 7.9 | 3.5 | 3.9 | 7.9 | 3.1 | 6.0 | 7.8 | 2.9 | 6.1 | 8.0 | 3.5 |

Table 6: MT-Bench scores for methods initialized from SFT-base trained setting. Ultra refers to UltraFeedback dataset, Edu refers to EduFeedback-alternate dataset, and IMDb refers to the seminal IMDb dataset.

| Method | Edu | IMDb | Ultra |
|---|---|---|---|
| COALA | **66.67** | **65.23** | **65.38** |
| ORPO | 46.67 | 45.46 | 53.33 |
| DPO | 61.33 | 52.17 | 57.14 |
| SFT | 6.67 | 37.50 | 25.00 |

Table 7: ArenaHard Custom Pairwise WR_ex_Ties: Dolphin-7B Win Rate Excluding Ties

# E  PREFERENCE ALIGNMENT METHODS AND DATASETS

## E.1  METHODS OVERVIEW

In this section we introduce the three preference alignment methods of our experiments. Table 8 explicitly compares the objective of each method, and its associated hyperparameters. Each method is evaluated against three preference datasets and five model architectures, for a total of **105** experimental runs.

**COALA Method.** All COALA experiments are executed on one RTX-4090 with 24GB VRAM. The base model to be fine-tuned is frozen, and *preference embeddings* are extracted. During stage one COALA training we use CRONOS to train a cvxNN as a preference classifier. During stage two COALA training we further fine-tune these weights with the COALA loss objective. Our JAX implementation takes full advantage of GPU acceleration and lifts memory constraints. Therefore even large models (such as LLaMA-8B) can be trained with COALA on one RTX-4090, much faster than DPO or ORPO methods with LORA and DeepSpeed (Rajbhandari et al., 2020) on the A100 GPU.

| Method | Objective | Hyperparameter |
|---|---|---|
| **RRHF** (Yuan et al., 2023) | $\max\left(0, -\frac{1}{|y_w|}\log\pi_\theta(y_w|x) + \frac{1}{|y_l|}\log\pi_\theta(y_l|x) - \lambda\log\pi_\theta(y_w|x)\right)$ | $\lambda \in \{0.1, 0.5, 1.0, 10.0\}$ |
| **DPO** | $-\log\sigma\left(\beta\log\frac{\pi_\theta(y_w|x)}{\pi_{\text{ref}}(y_w|x)} - \beta\log\frac{\pi_\theta(y_l|x)}{\pi_{\text{ref}}(y_l|x)}\right)$ | $\beta \in \{0.01, 0.05, 0.1\}$ |
| **ORPO** | $-\log p_\theta(y_w|x) - \lambda\log\sigma\left(\log\frac{p_\theta(y_w|x)}{1-p_\theta(y_w|x)} - \log\frac{p_\theta(y_l|x)}{1-p_\theta(y_l|x)}\right)$ | |
| | where $p_\theta(y|x) = \exp\frac{1}{|y|}\log\pi_\theta(y|x)$ | $\lambda \in \{0.1, 0.5, 1.0, 2.0\}$ |
| **R-DPO** | $-\log\sigma\left(\beta\log\frac{\pi_\theta(y_w|x)}{\pi_{\text{ref}}(y_w|x)} - \beta\log\frac{\pi_\theta(y_l|x)}{\pi_{\text{ref}}(y_l|x)} + (\alpha|y_w| - \alpha|y_l|)\right)$ | $\alpha \in \{0.05, 0.1, 0.5, 1.0\}, \beta \in \{0.01, 0.05, 0.1\}$ |
| **SimPO** | $-\log\sigma\left(\frac{\beta}{|y_w|}\log\pi_\theta(y_w|x) - \frac{\beta}{|y_l|}\log\pi_\theta(y_l|x) - \gamma\right)$ | $\beta \in \{2.0, 2.5\}, \gamma \in \{0.3, 0.5, 1.0, 1.2, 1.4, 1.6\}$ |
| **COALA** | $\log\left(1 + \exp\left(-\beta y_w\theta_2^T(\Theta_1 f_{\theta_{\text{pre}}}(x))_+ + \gamma\right)\right)$ | CRONOS parameter $\rho = 0.01, \gamma$ |

Table 8: Summary of popular preference fine-tuning methods, respective objective functions, and hyperparameters for completeness.

**DPO Method.** The traditional DPO Rafailov et al. (2024) algorithm requires a frozen reference model to stabilize training. Since this exceeds the VRAM constraints of our single GPU setting (especially for large models such as LLaMA-8B), we only experiment with offline DPO training on one A100 GPU with 40GB VRAM. We comprehensively run DPO with three datasets on ten base models, for a total of thirty resulting DPO checkpoints. All DPO training uses LORA (Hu et al., 2022) and DeepSpeed (Rajbhandari et al., 2020) for GPU acceleration in Python.

**ORPO Method.** ORPO (Hong et al., 2024) is a reference-free method that aims to generalize to unseen data while reducing biases such as response length exploitation. The addition of the log odds ratio term helps to stabilize reward margin increments, and reduces the dependence on initializing from a preference-aligned SFT base model. However, ORPO is the slowest among all methods to complete one epoch of training and typically requires 100x smaller learning rate thank SFT. We implement a total of thirty ORPO runs to compenhensively evaluate its performance.

### E.2 DATASETS

In this section we provide explicit details of training datasets used in all experiments. In the case of SFT training, general conversational datasets are utilized (EduFeedback, UltraFeedback, IMDb). In the case of preference alignment, training datasets are formatted into "prompt, chosen, rejected" triplets. This distinction is critical, since the curation of preference training triplet datasets for new domains is non-trivial. The current convention requires extracting the last "assistant response" from each single conversation, then utilizing external expensive LLMs to generate a "chosen" or "rejected" response that is slightly different. These implications are that for a conversation dataset of N convos with multiple turns, only N training triplets can be extracted for preference learning. In this paper we introduce the **Alternating Population Strategy**, which extracts the same number of preference training triplets as the number of **assistant** responses in the entire conversation dataset. We detail the effectiveness of our novel data extraction method below (EduFeedback-Alternate). This is motivated due to the fact that since data for machine learning is becoming increasingly scarce, it is critical to extract the maximum amount of value possible from available datasets.

- **EduFeedback** This SFT is our custom conversational dataset inspired by UltraChat (Ding et al., 2023), but clearly constrained in an educational setting. EduFeedback contains 26,621 conversations generated synthetically with GPT-4o (OpenAI, 2024). We vary $t = 0.2 - 0.9$, utterances range from 4-8 alternating turns between two agents. We randomly vary *mood* of agents to simulate realistic human conversation, and encompass eleven diverse topics in fields of study such as science and philosophy. This creates 26,621 diverse and realistic conversations between a student studying for a quiz, and an academic tutor assisting the student in a natural conversational environment.

- **EduFeedback-Alternate** This is the extracted version used for preference fine-tuning, and contains "chosen, rejected, prompt" triplets. Typical preference datasets are extracted from conversational formats by extracting the "prompt" as all but the final two turns in the conversation. The "chosen" and "rejected" selections are then populated by sampling from

the SFT baseline model around 5 times, and given to PairRM (Jiang et al., 2023b) (or a third party model) for a "preference score". This strategy is slow, expensive, and limits the number of preference training samples available. It is also sensitive to choice of various scoring models, or various methods of generating "chosen" and "rejected" responses. As a result the fidelity of the fine-tuned model shows a high variance in performance. Instead we propose the novel **Alternating Population Strategy** for generating preference datasets from conversations. By selecting the "prompt" in each conversation to be "agent 2", populating the "chosen" as the immediate next "agent 1" response $i$, populating the "rejected" as the $i+2$ response. We then proceed to populate the next "prompt" as the concatenation of the $i+1$ response with all previous responses, and continue. In this way a conversation of 3-4 turns will populate 2-3 training triplets of "prompt, chosen, rejected", thus resulting in 65,606 training samples in the custom EduFeedback-Alternate for preference alignment.

- **UltraFeedback-Binarized, (Face, 2023)** This benchmark dataset is selected to be consistent with prior work Meng et al. (2024). It features preselected training triplets for preference learning, and 60,917 samples. The highest scoring completion from UltraFeedback was selected as "chosen" with the lowest score selected as "rejected".

- **UltraFeedback, (Cui et al., 2023)** The original Ultrafeedback dataset comprises 64,000 training samples each containing four model-generated completions from a variety of open-source models. GPT-4 (OpenAI, 2023) was used to assign "preference scores" to each completion. This dataset serves as a standard most general equivalent benchmark.

- **IMDb, (Maas et al., 2011)** The IMDb positive-negative sentiment analysis dataset is selected to be consistent with prior work Rafailov et al. (2024). In each instance the "prompt" is either negative or positive, and generation quality is graded on the adherence and cogency of the of the output with regards to sentiment. We downsample this dataset to 11,000 training samples.

### E.3 ALTERNATING POPULATION STRATEGY

---

**Original Multi-turn Conversation**

**System:** You are Dolphin, a helpful AI tutor.

**User (Agent 2):** Hi! I'm studying for my music class and need some help understanding the different periods in classical music. Can you explain the main characteristics that differentiate the Baroque period from the Romantic period?

**Assistant (Agent 1, Response i):** Certainly! The Baroque period (roughly 1600-1750) is characterized by intricate compositions, ornate embellishments, and a strong emphasis on harmony and counterpoint. Music from this era often features complex polyphonic textures and uses forms such as the fugue.
In contrast, the Romantic period (roughly 1800-1910) focuses more on emotional expression and individuality. Music from this time often has expansive melodies, dynamic contrasts, and richly textured chords. Composers from this period were more inclined to experiment with form and structure, often seeking to evoke intense emotions and depict narratives or landscapes through their music.

**User (Agent 2):** I need more help to study for my music class tomorrow. What if a question on the quiz asks me to enumerate the main differences between the Baroque and Romantic periods in Classical music?

**Assistant (Agent 1, Response i+2):** During the Classical period (roughly 1750-1820), the role of the conductor was often filled by a leading musician within the ensemble, such as the first violinist or the keyboard player, who led the group mainly through visual cues and playing. There wasn't a separate, distinct role of a conductor as we see today. By the 19th century, as orchestras grew in size and the music became more complex, the need for a single individual to manage the performance became evident. This led to the emergence of the modern conductor who stands on a podium with a baton to lead the orchestra...

---

---

**Extracted Preference Triplet Training Sample**

**Prompt:**
```
<|im_start|>system
You are Dolphin, a helpful AI tutor.<|im_end|>
<|im_start|>user
agent2: Hi! I'm studying for my music class and need some help
understanding the different periods in classical music. Can you
explain the main characteristics that differentiate the Baroque
period from the Romantic period?<|im_end|>
```

**Chosen Response:**
```
<|im_start|>assistant
```
Certainly! The Baroque period (roughly 1600-1750) is characterized by intricate compositions, ornate embellishments, and a strong emphasis on harmony and counterpoint. Music from this era often features complex polyphonic textures and uses forms such as the fugue.
In contrast, the Romantic period (roughly 1800-1910) focuses more on emotional expression and individuality. Music from this time often has expansive melodies, dynamic contrasts, and richly textured chords...
```
<|im_end|>
```

**Rejected Response:**
```
<|im_start|>assistant
```
During the Classical period (roughly 1750-1820), the role of the conductor was often filled by a leading musician within the ensemble, such as the first violinist or the keyboard player, who led the group mainly through visual cues and playing...
```
<|im_end|>
```

---

**Key Advantages.** We introduce this novel strategy to generate preference data where the *chosen* response directly addresses the user's query about Baroque vs. Romantic periods, while the *rejected* response, though informative and relevant, discusses a less direct topic (Classical period conductors). This creates natural preference pairs without requiring external scoring models or external generative models, and provides an efficient yet effective method to populate $65\,606$ preference training samples from the $26\,621$ sample EduFeedback dataset. Traditional preference datasets are typically expensive to generate, and yield the same number of training-triplets (*prompt, chosen, rejected*) as conversations. This heuristic is specifically designed for objective contexts where the initial response to a user query ($i$) is typically a direct answer, while subsequent turns ($i+2$) often involve tangents or deeper follow-ups that are still relevant but less direct as a primary response. To validate this, our human evaluation (Appendix G) implicitly tests the quality of this data strategy: the high win-rates of COALA trained on this data suggest the heuristic successfully captures genuine preference signals in concise educational domains. Figure 3 also demonstrate the practical feasibly of this population strategy, as reward margins show stable improvement.

# F EXPERIMENTAL SETUP DETAILS

## F.1 HARDWARE CONFIGURATIONS

COALA is trained using a single RTX-4090 GPU (24GB) with JAX, while SFT, DPO, and ORPO use individual A100 GPUs (40GB) with PyTorch. To ensure fair comparison, each (Method × Dataset × Model) setup is run at least three times. Unlike other methods, COALA requires no PEFT tuning, and instead leverages CRONOS (Section 4) for efficient optimization over high-dimensional features for its stage one training. Table 9 explicitly details hardware setup across experiments, and the following subsection F.2 details PEFT configurations used for the strongest competitors.

## F.2 LORA AND DEEPSPEED CONFIGURATIONS FOR COMPETING METHODS

Table 11 Gives the precise DeepSpeed configurations used to achieve the most competitive results via SFT, DPO, and ORPO methods. Table 11 Gives the precise DeepSpeed configurations used to achieve the most competitive results via SFT, DPO, and ORPO methods.

| Method | GPU | VRAM | Acceleration Method |
|--------|-----|------|---------------------|
| COALA | RTX 4090 | 24GB | JAX with XLA |
| DPO | A100 | 40GB | DeepSpeed + LoRA |
| SFT | A100 | 40GB | DeepSpeed + LoRA |
| ORPO | A100 | 40GB | DeepSpeed + LoRA |

Table 9: Hardware and framework settings used across experiments.

| Parameter | Value |
|-----------|-------|
| r | 16 (up to 256 in DPO setting) |
| lora_alpha | 32 (up to 128 in DPO setting) |
| lora_dropout | 0.05 |
| bias | none |
| task_type | CAUSAL_LM |
| target_modules (GPT-2 base) | [c_attn, c_proj] |
| target_modules (all other models) | [q_proj, k_proj, v_proj, o_proj, gate_proj, up_proj, down_proj] |

Table 10: LoRA Configuration Settings for Competing Methods

## F.3 INFERENCE DETAILS

Our guided sampling pipeline integrates the convex model directly into generation. At each step, we use prefix-based nucleus sampling (`top-p`=0.9, `top-k`=50, `num_candidates`=5) to generate candidates, score them using a contrastive objective and select the highest-scoring continuation. While both prefix-based and token-by-token modes are supported in our inference module, COALA experiments specifically utilized prefix-based. We emphasize that COALA's convex program is integrated into this structured guided sampling framework. rather than beam search. Although COALA targets single-GPU resource constraints, our modular JAX inference pipeline is written to scale to 8xA100 GPU environments, and supports low-memory devices via device-aware model loading. We are excited to release the sampling code alongside the training codebase, and have included a section in the Appendix to show the generative pipeline in the revision. Since COALA is designed and presented as a novel convex approach training algorithm for preference alignment of LLMs, in order to be consistent with prior work such as DPO and SimPO, we focus primarily on the training stage in this work. Our benchmarks and ablations target training effectiveness and compute efficiency. Inference-time generation strategies are significant promising directions, which we look forward to in future work.

## G HUMAN PREFERENCE AND EVALUATION DETAILS

We comprehensively validate the quality of generated output with 107 double blind real human samples. This is consistent with the previous seminal work by Rafailov et al. (2024), and results agree with our statistical findings in this paper. Survey is conducted via 25 multiple choice questions per category in the same length and format as Table 4. Each human participant is asked to select the most **positive** movie review (for the IMDb dataset task) or the most **helpful** answer to an educational question (for the EduFeedback dataset task). Each multiple choice answer is generated from a SFT-based trained LLaMA-8B model preference aligned with COALA, DPO, ORPO, or baseline SFT. Subsections G.1 and G.2 provide samples of survey questions with associated percentage human win-rate per method, subsection G.3 details the consent form each person signed to ethically participate in the study, and Table 2 in the main work summarizes numerical results. In order to adhere to the double blind process during reviews we temporarily omit the names of the human samples during submission.

Table 11: DeepSpeed configuration used for training competing methods.

| Parameter | Value |
|---|---|
| *BF16* | |
| bf16.enabled | true |
| *Optimizer (AdamW)* | |
| optimizer.lr | $2.0 \times 10^{-4}$ |
| optimizer.betas | [0.9, 0.999] |
| optimizer.eps | $1 \times 10^{-8}$ |
| optimizer.weight_decay | 0.01 |
| *Scheduler (WarmupDecayLR)* | |
| scheduler.warmup_min_lr | 0 |
| scheduler.warmup_max_lr | $2.0 \times 10^{-4}$ |
| scheduler.warmup_num_steps | 1 000 |
| scheduler.total_num_steps | auto |
| *Zero Optimization (Stage 3)* | |
| zero_optimization.stage | 3 |
| zero_optimization.offload_optimizer.device | none |
| zero_optimization.offload_optimizer.pin_memory | true |
| zero_optimization.offload_param.device | none |
| zero_optimization.offload_param.pin_memory | true |
| zero_optimization.overlap_comm | true |
| zero_optimization.contiguous_gradients | true |
| zero_optimization.sub_group_size | $1 \times 10^9$ |
| zero_optimization.reduce_bucket_size | $3 \times 10^8$ |
| zero_optimization.stage3_prefetch_bucket_size | $3 \times 10^8$ |
| zero_optimization.stage3_param_persistence_threshold | $3 \times 10^8$ |
| zero_optimization.stage3_max_live_parameters | $1 \times 10^9$ |
| zero_optimization.stage3_max_reuse_distance | $1 \times 10^9$ |
| zero_optimization.stage3_gather_16bit_weights_on_model_save | true |
| gradient_accumulation_steps | 8 |
| gradient_clipping | 1.0 |
| train_micro_batch_size_per_gpu | 2 |
| train_batch_size | 16 |
| wall_clock_breakdown | true |

### G.1 EDUFEEDBACK SURVEY QUESTIONS AND EXAMPLES OF RESULTS

**What is the most valuable work of art?**

- SFT (20.6%): "The most valuable work of art is the portrait of Leonardo da Vinci. Leonardo created the p..."
- ORPO (9.3%): "The most valuable work of art is the human heart, which is the most important of all objects i..."
- COALA (30.8%): "The value of a work of art can be measured in various ways,..."
- DPO (39.3%): "The value of art is subjective, but one artwork that is often c..."

**What is a sonata?**

- SFT: "A sonata is a piece of music composed by a child. Each sonata is represented by a dif..."
- ORPO: "Sonata is a Greek word meaning b̈eautiful, light, and joyful.S̈onata provides a wi..."

- COALA (65.4%): "A sonata is a form of musical composition that typically con..."
- DPO (29.9%): "The sonata (Italian, s̈ounding) is a piece of mu..."

**Define the Enlightenment period.**

- SFT (7.5%): "The Enlightenment period was marked by a significant period of intellectual and political dev..."
- ORPO (15.0%): "The Enlightenment period is a period in the human history, during which humanity's most..."
- COALA (47.7%): "The Enlightenment was an intellectual and philosophical..."
- DPO (29.9%): "The Enlightenment period is a philosophical movement whic..."

**What does glucose do?**

- SFT (9.3%): "Glucose is a critical component of human metabolism, which helps break down fat and car..."
- ORPO (9.3%): "Glucose is a glucose-containing carbohydrate found in the blood and in the body. It's a key co..."
- COALA (57.9%): "Glucose is a type of sugar that our bodies use for energy. It's..."
- DPO (23.4%): "Glucose is a monosaccharide that is the main source of ene..."

**What is surrealism in art?**

- SFT (17.8%): "In art, surrealism is a form of expression, with a focus on dramatic elements, often in ju..."
- ORPO: "In art, it can be difficult to appreciate the beauty of a piece. Often, the artist's intent..."
- COALA (65.4%): "Surrealism is an artistic movement that emerged in th..."
- DPO (10.3%): "Surrealism is a cultural movement, which was founde..."

**What is the theory of Carl Jung?**

- SFT (14.0%): "Carl Jung was a Swiss psychologist who proposed the theory of archetypes, which a..."
- ORPO (9.3%): "The theory of Jung, which is the theory of Jung's theory of the process of mental developme..."
- COALA (57.0%): "Carl Gustav Jung (1875-1961) was a Swiss psychiatrist and..."
- DPO (19.6%): "Carl Jung was a Swiss psychiatrist who founded anal..."

**What was the Cold War?**

- SFT (20.6%): "The Cold War was a period of geopolitical tension between the Soviet Union and the United..."
- ORPO (24.3%): "The Cold War was an era during which the United States and its allies waged a war ag..."
- COALA (51.4%): "The Cold War was a period of geopolitical tension between t..."
- DPO: "The Cold War was a period of tension between the United S..."

G.2 IMDb SURVEY QUESTIONS AND EXAMPLES OF RESULTS

**The movie was so...**

- SFT (15.0%): "intense about a young man, M̈ikeyẅho is the child of an alcoholic mother. His mom di..."

- ORPO (9.3%): "wonderful to hear that you enjoyed the movie! I'm glad that you found it to be a unique an..."
- COALA (57.0%): "awesome. It was the best movie I have seen in a while. The ac..."
- DPO (18.7%): "incredible! I loved it so much it's not even funny. I'm the type w..."

**I went to see the movie and thought it was...**

- SFT: "a Friday night and I got there early, so I was able to get a great seat. The movie was pl..."
- ORPO (18.7%): "great and I enjoyed the movie! I'm sorry that the movie left me with mixed emotions. It's und..."
- COALA (66.4%): "so good I saw it twice in the theatre. I know there are a lot..."
- DPO (14.0%): "good! I loved it! My dad is a big fan of the old movie and the s..."

**After watching the film, I felt...**

- SFT (21.5%): "it's still one of my all time favorite movies. The movie was so unbelievable that you were..."
- ORPO (9.3%): "the movie is great! I love the way the movie plays with kids. It's interesting to see how the..."
- COALA (38.3%): "that it was very well done. I felt that the acting was very good...."
- DPO (30.8%): "a real sense of satisfaction. The film is, for me, a classic. The..."

**The director's vision for this movie...**

- SFT (8.4%): "is a good way to start a review, and I was not disappointed with this film. I was expecting so m..."
- ORPO (57.0%): "was to create a thought-provoking and emotionally engaging narrative that explor..."
- COALA (34.6%): "is very clear and it is well executed, the movie is very b..."
- DPO: "is something that I've always wanted to see for a long time,..."

**The actors in this movie performed...**

- SFT (10.3%): "gritty, hard-hitting, and unflinchingly honest looks at the world. The film follows the live..."
- ORPO: "great! I can't even begin to express how awesome they were. The actors are all very f..."
- COALA (43.9%): "admirably and believably, the characters were well-rounded..."
- DPO (40.2%): "to perfection. In this movie, the characters are really the stars..."

**The cinematography in this film was...**

- SFT (18.7%): "amazing. The film starts out with a woman in a room, and as the camera pans out, we see..."
- ORPO (26.2%): "pretty thrilling, about a group of friends who are trapped in a haunted house and must find..."
- COALA (29.9%): "beautiful. It had a great feel to it, and was very realistic, which..."
- DPO (25.2%): "incredible. It was shot like a documentary, but with a lot of..."

**My favorite scene in the movie was when...**

- SFT: "Hicks and Bishop are trying to kill the alien in the corridor, while a woman is screaming ¿.."
- ORPO (15.0%): "through the use of stunning visuals, powerful performance..."
- COALA (30.8%): "Alice is walking through the forest looking for Dave and sh..."
- DPO (28.0%): "the main character, played by the fabulous and talented Jud..."

### G.3 HUMAN EVALUATION CONSENT FORM

**Title of Study:** Human Evaluation of Preference-Aligned Language Model Show Down

**Principal Investigator:** Omitted for submission.

You are being asked to participate in a research study that evaluates human preferences for different fine-tuning algorithms used to align large language models (LLMs), such as COALA, DPO, ORPO, SimPO, CPO, GRPO and SFT. Your responses will help assess the quality and alignment characteristics of LLM-generated outputs.

**What Participation Involves:**

- You will complete a survey consisting of multiple-choice questions based on outputs from various LLM alignment methods.
- You will be asked to select responses based on perceived **correctness**, **helpfulness**, and **humanness**.

**Use of Data and Publication:**
By signing this form, you agree to the following:

- Your responses may be used in published research, academic papers, and open-access benchmarks evaluating LLM alignment.
- The results may be shared online, including datasets and analysis.
- Your name and institutional affiliation will be publicly disclosed as part of the human evaluation dataset, for transparency and attribution.

**Confidentiality & Voluntary Participation:**
Participation is voluntary. You may withdraw at any time before submission. After submission, your data—including your identity—may be published as part of a public benchmark and cannot be withdrawn.

**Acknowledgment & Consent:**
By signing below, you confirm that:

- You have read and understood the purpose and procedures of this study.
- You understand that your responses, name, and affiliation will be publicly shared as part of the dataset and may appear in future publications.
- You consent to participate in this research under these terms.

Participant Name: _________________________________

Institutional Affiliation: _________________________________

Email (optional): _________________________________

Signature: _________________________________

Date: _________________________________

# H    ADDITONAL STATEMENTS AND DISCUSSION

**Ethics Statement.** Prior to inviting our human samples to participate in the "LM Fine-tuning Show Down Survey", we explicitly collected consent via an example of the G.3. During submission we have removed certain institutional details in order to adhere to the double-blind process.

**Reproducibility statement.** In addition providing the JAX and modular package for reproducing experiments, detailed parameters and GPU set-ups are reported in Appendix F.

