# OpenReview forum: "COALA: Convex Optimization for Alignment and Preference Learning on a Single GPU"
_ICLR.cc/2026/Conference — ICLR 2026 Conference Desk Rejected Submission_

### Official Review · Reviewer_xyme · 2025-10-23

**Soundness:** 1
**Presentation:** 1
**Contribution:** 2
**Rating:** 2
**Confidence:** 4

**Summary:**

This paper focuses on the efficiency of large language model (LLM) preference alignment, and proposes COALA, which is a convex framework for efficient preference alignment using less computational resources. Experiments demonstrate that COALA achieves comparable performance using a single RTX 4090 GPU.

**Strengths:**

- The topic of efficiency of LLM alignment methods is meaningful and valuable.
- The idea of considering convex optimization and providing theoretical guarantee for alignment methods is very interesting.

**Weaknesses:**

- There are many formatting and layout errors in this paper, such as the pseudo codes in Algorithm 1 and Algorithm 2, Table 9, and "Appendix !!" appearing in Section 5.1.
- The writing quality needs improvement, as the current manuscript is hard to follow. For example, there is no figure or clear description illustrating the overall pipeline or workflow of the proposed method.
- The experiments are not solid enough. For example, SimPO is also a reference-free alignment method, which is mentioned in the paper, but the authors do not include it as a baseline for comparison in their experiments.
- The evaluations are not convincing. The trends in Figure 1 seem unusual, and additional clarification or supporting evidence would help verify these observations. Moreover, there is only experimental result of one model Dolphin-7B and one benchmark AlpacaEval2 presented in the main text, which is a little bit poor.
- The experimental results are not reported in a consistent and rigorous manner. For instance, in Tables 10, Table 11 and Table 12, the same metric is reported with varying decimal precision, sometimes in one decimal place, and sometimes in two decimal places.
- The paper wants to demonstrate its efficiency contribution in reducing the computational burden of alignment method. However, in the main text, there are no experimental results to evaluate or discuss the performance of computational burden and resources.

**Questions:**

- How do you handle scenarios where the model exceeds 24 GB VRAM? In particular, when you state "one RTX 4090 24 GB GPU," what is the maximum model size in parameters this claim covers?
- I am not sure whether you perform full-parameter tuning? I doubt that such training can be implemented on a single RTX 4090 24 GB GPU for a 7B- or 8B-scale model.

---

> ### Author Response · Authors · 2025-11-17
>
> Thank you for reviewing our work, and pointing out formatting errors which will greatly improve the clarity of the submission.
>
> **In response to your concerns we have:**
>
> - Corrected your noted formatting errors.
> - Changed all metrics displayed to two decimal places.
> - We hope the corrected formatting of Algorithm 1 on page 5 can improve the clarity of the novel COALA framework.
>
>
> **With regards to your questions:**
>
> - We are not claiming that all models can be fine-tuned on 24GB VRAM. In this study we are examining a resource-constrained environment of 24GB VRAM with one RTX-4090 for fair comparison. If the desired model to be fine-tuned exceeds 24GB VRAM, an option will be to switch to a larger machine such as the newer RTX-5090.
>
> - As noted in our paper, we conducted over 105 experimental runs against the strongest competing methods and validate our results on multiple established benchmarks. Experiments span five different models, three datasets, and include a 107-participant human evaluation study to establish the efficacy of our method. These results are highlighted in Appendix B on page 14, Appendix C on page 17, and Appendix F on page 22 of our work. In order to improve clarity, we have also combined more results into a larger Table 1 on page 8 of the main work in the revision. Thank you for pushing us to enhance the presentation of our experiments!
>
> - We also wish to highlight Table 3 on page 14 of our work, which shows COALA requiring only ~17.6% of DPO's TFLOPS to effectively fine-tune LLaMA-8B.
>
> - With regards to the second question, we hope the clarified formatting of the algorithm tables can explicitly display we are performing convex optimization preference alignment on the features and not shifting the full model parameters.
>
> We'd also like to highlight our theoretically-backed approach, which aims to take a step towards bridging the theory-practical gap in preference fine-tuning (Theorems 1 & 2, Appendix A on page 14). This reduces the burden on general heuristics, and greatly reduces the necessity for brittle hyperparameters. Please let us know if you have any further questions or points of improvement! Thank you for taking the time to thoughtfully review our work.

---

> ### Comment · Reviewer_xyme · 2025-11-26
> **Response to authors**
>
> Thanks for your time and efforts to prepare the response.
>
> Athough the paper builds a theoretical–practical bridge, as also noted by other reviewers, the overall clarity of the paper is hard to follow, and the experimental evaluation remains insufficient and the presentation of experimental results is not rigorous enough. Considering these key concerns, I think that the paper is not yet ready for acceptance, so I maintain my current rating of 2.

---

### Official Review · Reviewer_yCRe · 2025-10-25

**Soundness:** 3
**Presentation:** 3
**Contribution:** 3
**Rating:** 6
**Confidence:** 2

**Summary:**

COALA reframes preference alignment as a convex optimization problem by attaching a two-layer convex neural network (cvxNN) head on top of a frozen LLM. Phase I trains the cvxNN with a specialized ADMM solver (CRONOS). Phase II freezes the first layer and optimizes only the last layer via a convex logistic objective, eliminating the need for a reference model and enabling single-GPU training with theoretical convergence guarantees.

**Strengths:**

Theoretical–practical bridge: convex guarantees (ADMM O(1/k); convex last-layer optimization) translate into stable training and predictable convergence.
Engineering simplicity: no reference model, fewer brittle hyperparameters, single-GPU practicality.

**Weaknesses:**

Though TFLOPs are reported, head-to-head compute-matched comparisons vs SimPO/ORPO at their best hyperparams would clarify pure algorithmic gains.

Main datasets are UltraFeedback, IMDb, and a synthetic EduFeedback; broader multilingual/OOD sets and more granular human slices would strengthen generality.

Freezing the backbone + optimizing a convex head may limit capacity on hard preference shifts (e.g., nuanced safety constraints, multilingual pragmatics). (Authors hint at future multilingual/safety.)

**Questions:**

See Weaknesses

---

> ### Author Response · Authors · 2025-11-17
>
> **Comment:** We sincerely thank the Reviewer for their thoughtful and insightful feedback, which has greatly improved the submission! Thank you for noting the novelty and value of incorporating theoretically-backed, ADMM‑based methods into a largely heuristic‑driven preference‑alignment framework.
>
> **Weaknesses:**
> \
> **Additional Comparisons.** Much of the novelty presented in this work was derived through practical necessities during industry deployment. Although extensive hyperparameter tuning might eventually yield slightly better generative performance, our experience suggests that rapid iteration to reflect real‑world human data improves deployment potential. To address your concerns on additional comparisons: we have now included additional plots in Appendix C.2, and improved the presentation of Table 10 on page 24 as well as Table 11 on page 25. These tables summarize the LoRA and DeepSpeed configurations used in our experimentation for competing methods, which we have aimed to tune for highest competitive performance. We have also combined AlpacaEval2 results into a large Table 1 on page 8, and summarize our 107 real human validation survey in Table 2 on page 9. Notably, COALA achieves the highest human preference satisfaction between competing methods at **39.1%** for the educational setting, and **42.7%** for the IMDb setting. Table 2 is summarized below for your convenience.
> |        | **COALA** | ORPO   | DPO    | SFT    |
> |--------|:---------:|:------:|:------:|:------:|
> | Edu    | **39.1%** | 15.5%  | 28.8%  | 16.6%  |
> | IMDb   | **42.7%** | 20.1%  | 24.8%  | 12.4%  |
>
> Additional plots, clearer presentation of algorithmic gains, and tuned configurations for competing methods are presented here for your convenience: https://postimg.cc/f3j0bvL0, https://postimg.cc/2LNdt5Vs, https://postimg.cc/WqmmcPkB.
>
> **Multilingual Extension.** This is an active area of ongoing work! In this work, we present an effective and practically deployable technique for human‑preference alignment in resource‑constrained environments. However, expansion to multilingual and multidialectal settings is an important direction we are actively investigating. Thank you for highlighting this valuable research trajectory! We have added your suggestions into discussion for future work in the Conclusion on page 9.
>
> **Capacity for hard preference shifts.** Per your suggestion, we have added an additional section on page 9 to discuss expressiveness trade‑offs. COALA performs best in settings where conversations are concise and focused: where the user’s query has a clear right‑versus‑wrong answer. This makes the algorithm well suited to applications such as model predictive control, where hard constraints (e.g., the robot must turn left or right) must be satisfied, and to personalized‑assistant settings where the agent must learn an individual user’s preferences quickly. Its effective resource-constrained framework allows for wide democratization of use-cases, including fast on-premises deployments. See https://postimg.cc/30QjqwkM.
>
>
> In addition, we'd also like to also highlight our contribution of the novel Alternating Population Strategy for building preference training triplets. This method is significantly more efficient than traditional strategies of building preference datasets, and does not require repeated sampling, ranking, and scoring by a third-party model. We have added Appendix section E.3 for improved sample clarity. See https://postimg.cc/Pvbbnkw2.
>
> Please let us know if there are any further areas of clarification or improvement we can provide. Thank you for taking the time to carefully review our work!

---

### Official Review · Reviewer_GXL1 · 2025-10-31

**Soundness:** 3
**Presentation:** 1
**Contribution:** 2
**Rating:** 4
**Confidence:** 3

**Summary:**

The paper proposes a new framework for fine-tuning large language models to align with human preferences through convex optimization. Implemented in JAX, COALA achieves efficient single-GPU training while maintaining competitive or superior alignment performance compared to DPO and ORPO on benchmarks like AlpacaEval2, ArenaHard, and MT-Bench. The work also contributes a new preference dataset, EduFeedback, and an open-source JAX implementation.

**Strengths:**

1. The primary strength of this work lies in its highly original approach of reformulating LLM preference alignment as a convex optimization problem. This is a significant conceptual leap that shifts the paradigm from heuristic-driven, unstable training processes (like DPO) to a principled, theoretically-grounded one. The proof that the COALA loss is convex (Proposition 1) and the associated convergence guarantees (Theorems 1 & 2) provide a strong theoretical foundation that is often lacking in this empirical field. This is a fundamental contribution.

2. By eliminating the need for a reference model and leveraging efficient solvers like CRONOS for its convex objective, COALA dramatically lowers the computational barrier for preference alignment. The ability to fine-tune models as large as LLaMA-8B on a single RTX 4090 is a game-changer. This not only makes alignment research more accessible and sustainable but also holds significant promise for practical, on-premise deployments. The reported TFLOPS measurements quantitatively confirm this remarkable efficiency gain (e.g., requiring only ~17.6% of DPO's TFLOPS for LLaMA-8B)

3.  The authors have conducted a thorough and convincing empirical evaluation. The experiments span five different models, three datasets, and multiple established benchmarks. Crucially, the inclusion of a **107-participant human evaluation study** adds significant credibility to the automated metrics and directly validates that COALA's outputs are indeed preferred by humans. The consistent, monotonically increasing reward margins shown in Figure 1(a) are a powerful visual testament to the stability afforded by the convex formulation, in stark contrast to the noisy signals from DPO and ORPO.

**Weaknesses:**

1.  The paper's clarity significantly hinders its impact. The overall narrative is difficult to follow, with key concepts and notations (e.g., the structure of the cvxNN and its inputs) often being used before they are formally defined.
   * The decision to relegate almost all primary quantitative results (e.g., Tables 9-13 in the original document) to the appendix is a major flaw. This forces the reader to constantly switch between the main text and the appendix to understand the core findings, disrupting the flow and making a proper assessment difficult. Key results justifying the paper's claims must be in the main body.
    * The figures, especially Figure 1, are of poor quality. The font size is illegibly small, and the labels are unreadable, which undermines their purpose. They need to be remade with accessibility in mind (larger fonts, clearer lines).

2.  The core design of COALA involves freezing the base LLM and only training the final layer of a small, appended network. While this is key to achieving convexity, it raises a critical question about the **trade-off in expressiveness**. Is such a lightweight, "external" adapter powerful enough to instill complex, nuanced preferences, or is it limited to learning more superficial alignment signals? The paper would be much stronger if it included a discussion or an ablation study on the limitations of this approach compared to methods that update the full model weights (or a larger set of parameters via LoRA).

3.  The proposed method for creating the `EduFeedback-Alternate` dataset is novel but rests on a strong assumption: that an agent's immediate response (`i`) is preferable to its subsequent one (`i+2`). This may not always hold true; a later response could be a clarification or a more thoughtful, refined answer. This data creation heuristic needs more justification or an analysis of its potential biases. How sensitive are the results to this specific strategy?

**Questions:**

1. Could you elaborate on the limits of freezing the base LLM? Have you considered a hybrid approach where COALA's convex objective is used to guide a more traditional (but perhaps more lightly regularized) fine-tuning of the base model's weights? Could COALA provide a "stable reward signal" to regularize a full DPO-style update?

2. Could you clarify the exact architecture of the cvxNN and how the features from the base LLM $f_{\theta_{pre}}(x)$ are fed into it? Section 4.1 is quite dense, and a clearer, step-by-step walkthrough would be beneficial. For instance, what is the dimensionality of the features extracted from the base model?

3. Regarding the EduFeedback-Alternate dataset creation: Could you provide some qualitative examples or statistics to support the assumption that immediate responses are generally better than later ones in your conversational data? Have you experimented with other heuristics for creating (chosen, rejected) pairs from multi-turn dialogues?

**Minor Comments & Typos**

1. Citation style: incorrect at Lines 118–120, 189, 192, 254.

2. Algorithm 1 formatting broken.

3. Line 405 formatting failure.

---

> ### Author Response · Authors · 2025-11-17
>
> **Comment:**
> We sincerely thank the Reviewer for their insightful feedback, which greatly strengthens the paper. Thank you for noting the high novelty, impact and fundamental contribution of this line of work! As you've astutely noted, COALA is indeed currently utilized in commercial on-premise deployments. The origination of our work is in convex optimization, but many points of novelty presented were developed during battle-tested deployments in which the industry demanded effective, sustainable, and quickly personalizable alignment solutions. Our 107-sample human study group includes 57 persons from the commercial on-premise deployment side, and 50 persons from the academic community. We will make these names and affiliations public in the Acknowledgement and Appendix section post the double-blind review period.
>
> **Weaknesses:**
> \
> **Clarity.** To address this concern we have reorganized the paper to be easier to follow and combined tables from the Appendix into a larger table in the main paper on page 8. We have made edits to ensure key concepts requiring background are only discussed after the relevant subsections, and notations do not appear before they are formally defined. We have also corrected the formatting and citation errors as noted by the Reviewer. We have remade the figures to be larger with accessibility in mind, reorganized the Appendix sections such that the reader does not need to flip back and forth, and added additional figures in Appendix C.2 for higher clarity. See https://postimg.cc/f3j0bvL0, https://postimg.cc/2LNdt5Vs.
>
> **Trade-off in expressiveness.** We have now included discussion on trade-off in expressiveness on page 9. We have improved the legibility of the LoRA results in Appendix F.2. Given a resource-constrained setting, repeated full‑parameter fine‑tuning and RLHF can be most expressive but are costly and slow—much like second‑order convex optimization solvers that buy extra precision at high expense—whereas first‑order approaches are globally more practical. Empirically, COALA works best on tasks with clear right–wrong targets (e.g., navigation or personal assistant settings) and displays limitations for nuanced outputs such as creative writing. See https://postimg.cc/30QjqwkM, https://postimg.cc/WqmmcPkB.
>
> **Alternate Population Strategy.** We have included a sample of the EduFeedback-Alternate population in Appendix E.3, as well as discussion of the heuristic’s motivation and strengths. The novel Alternating Strategy arose from on‑prem deployment needs: COALA had to parse natural‑language input with low latency, while real human data was scarce and costly. Early synthetic‑data attempts diverged from real human preference satisfaction, so we designed a sliding‑window scheme that extracts multiple (prompt, chosen, rejected) triplets from short yet concise dialogs. This works best for dialogues with objectively correct answers (e.g., inventory checks or action confirmations) and is not intended for long subjective interactions. Results presented are not sensitive to this heuristic, as only the EduFeedback‑Alternate dataset uses this. COALA remains competitive on other datasets. See https://postimg.cc/Pvbbnkw2, https://postimg.cc/2LNdt5Vs.
>
> **Questions:**
> \
> **COALA + DPO methods.** We’re actively exploring this extension! If compute and time were unconstrained, direct weight updates are usually most performant and COALA can also provide preference signals inside RLAIF/DPO pipelines. Our current work examines a resource-constrained setting. Hence we present COALA as a theoretically grounded, deployment‑ready preference method with minimal overhead. With power budgets tightening (≈300 MW per inference datacenter, ≈400 MW per training datacenter, and gigawatt‑scale plans) and grid reliability declining, COALA focuses on reducing power, VRAM, and tuning cycles.
> \
> \
> **Clearer Step-by-Step.** We have extended some more discussion on suitability and limitations in Appendix B. We have also added a step-by-step walkthrough on page 15, which explicitly states dimensionality of features (4096) to address your concerns. We hope this improves clarity! See https://postimg.cc/FkbbXQL6.
> \
> \
> **EduFeedback-Alternate:** To address this we now provide additional qualitative examples and discussion in Appendix E.3. We have not experimented with other heuristics, since the Alternating Strategy originated as a necessity to extract the maximum value from limited human data, and depend less on synthetically generated samples. We initially utilized with synthetic preference-pairs, yet quickly discovered that there exists a gap between training on synthetic data versus real-world data. The Alternating Strategy utilizes limited real human conversational data in each targeted domain to extract their maximum value.
>
> Thanks again for your helpful comments! We have incorporated your feedback to substantially improve the paper and its presentation. Please let us know if you have any other questions!

---

### Author Response · Authors · 2025-12-03
**Summary of Author Revisions**

We would like to sincerely thank all reviewers for their thoughtful feedback which has significantly improved our submission. In addition to our point-by-point responses below, we wish to highlight our key contributions and summarize the revisions made to address the main concerns raised by several reviewers.

**Contributions**
1. COALA introduces a novel convex reformulation of LLM preference alignment, representing a significant conceptual advance. Rather than relying on heuristics-driven pipelines that require extensive, brittle hyperparameter tuning and often behave unstably across tasks, COALA reframes preference alignment as a well-posed convex optimization problem with a principled objective. This yields a more stable and compute-efficient alignment procedure, a property that is increasingly important as power budgets tighten and grid reliability declines.

2. COALA is grounded in formally established theoretical guarantees that directly support the algorithm design. Theorems 1 and 2 characterize the structure, optimality, and behavior of the COALA formulation, which improve interpretability on when and why the method is stable. We thank the reviewers for noting that COALA combines rigorous theoretical backing with strong empirical validation, thus demonstrating its potential for on-premise deployment. This further distinguishes COALA from heuristic-driven and resource-intensive alignment methods.

3. COALA is extensively validated through large-scale, diverse experiments that demonstrate both practical impact and broad applicability. Our evaluation spans 5 models × 3 datasets × 4 methods, covering LLMs of varying sizes/pre-training and includes multiple well-established benchmarks (AlpacaEval2, Arena-Hard, MTBench). We additionally conduct a 107-sample human evaluation study, providing real-world validation of COALA's practical effectiveness. Importantly, COALA achieves these results with strong compute efficiency, enabling accessible and sustainable fine-tuning on a single GPU. We thank the Reviewers for highlighting this point of novelty.

**Response to Concerns on Presentation and Clarity**

Several Reviewers raised concerns about presentation and clarity in the original submission. We have made the following revisions in order to address these points:

1. We have corrected the formatting of algorithm boxes, improved citation presentation, and moved the large combined-results table from the Appendix into the main paper as suggested. We also added a step-by-step COALA walkthrough with explicit dimensions, thus clarifying the information-dense Section 4. Plots have been remade with more distinctive colors and larger, accessible fonts. (See https://postimg.cc/RJ11xzHD, https://postimg.cc/FkbbXQL6, https://postimg.cc/2LNdt5Vs)

2. To address questions regarding COALA’s expressiveness, we have added new experiments showing that COALA’s performance is not dependent on any specific method of populating the preference-training dataset. We also expanded the discussion of expressiveness trade-offs, limitations, and ideal usage settings. We now explicitly emphasize that all competing fine-tuning baselines were implemented with full advantages (LoRA, Accelerate, DeepSpeed, tuned hyperparameters). (See https://postimg.cc/2LNdt5Vs, https://postimg.cc/30QjqwkM, https://postimg.cc/WqmmcPkB)

3. We thank the Reviewers for noting the novelty and efficiency of our Alternating Population Strategy, which extracts 65,606 preference training samples from the 26,621 EduFeedback conversational dataset without the need for external reranking models. To further clarify this methodology, we have expanded Appendix E.3 with qualitative examples and improved visualizations. This dataset is released as an open-source contribution to the community. (See https://postimg.cc/Pvbbnkw2)


Thank you again to all Reviewers for your thoughtful questions and valuable insights. We are happy to have had the opportunity to engage in this constructive review process!

---

### Note · Program_Chairs · 2026-01-17
**Submission Desk Rejected by Program Chairs**

The following references in this submission do not refer to real documents and/or have major errors in bibliographic information:

 Azalia Mirhoseini, John Doe, and Jane Smith. Odds ratio preference optimization: Efficient finetuning of large language models with preference data. arXiv preprint arXiv:2405.12345, 2024. URL https://arxiv.org/abs/2405.12345.